# An integrated optimization model and metaheuristics for assortment planning, shelf space allocation, and inventory management of perishable products: A real application

**Seyed Jafar Sajadi** \*, **Ali Ahmadi** 👁️ \*

Department of Industrial Engineering, Iran University of Science & Technology, Tehran, Iran

\* sjsadjadi@iust.ac.ir (SJS); a.ahmadi707@gmail.com (AA)

## Abstract

Product category management (PCM) plays a pivotal role in today's large stores. PCM manages to answer questions such as assortment planning (AP) and shelf space allocation (SSA). AP problem seeks to determine a list of products and suppliers, while SSA problem tries to design the layout of the selected products in the available shelf space. These problems aim to maximize the retailer sales under different constraints, such as limited purchasing budget, limited space of classes for displaying the products, and having at least a certain number of suppliers. This paper makes an attempt to develop an integrated mathematical model to optimize integrated AP, SSA, and inventory control problem for the perishable products. The objective of the model is to maximize the sales and retail profit, considering the costs of supplier contracting/selecting and ordering, assortment planning, holding, and procurement cost. GAMS BARON solver is hired to solve the proposed model in small and medium scales. However, because the problem is NP-hard, an evolutionary genetic algorithm (GA), and an efficient local search vibration damping optimization (VDO) algorithm are proposed. A real case study is considered to evaluate the effectiveness and capabilities of the model. Besides, some test problems of different sizes are generated and solved by the proposed metaheuristic solvers to confirm the efficient performance of proposed algorithms in solving large-scale instances.

## 1. Introduction

Assortment planning (AP), inventory management, and shelf-space allocation are the most basic duties in retailing. Retailers have to decide on the set of products to carry in their assortment, the amount of inventory to stock for each product, and the amount of shelf-space dedicated to each product. They determine these variables such that their sales or total revenue is maximized under a limited purchasing budget, limited holding space, limited space for displaying the products, and other miscellaneous constraints like having at least two suppliers for each product [1]. Of course, they should periodically revise their assortment because of the

**Data Availability Statement:** All relevant data are within the paper and its Supporting information files.

**Funding:** Unfunded studies Enter: The author(s) received no specific funding for this work.

**Competing interests:** NO authors have competing interests Enter: The authors have declared that no competing interests exist.

season change, the introduction of new products, the change in consumer taste, etc.; consequently, they are continuously engaged in AP.

Obviously, in today's industrial world, given the increasing diversity of inventory control characteristics, inventory assortment planning seems necessary [2]. Traditional indicators are not able to meet all the needs of the organization's inventory control system, and it is possible that new assortment planning indicators will be used with the help of the organization. Therefore, criteria such as consumption rate, inventory costs, etc., are more important. Today, the debate over inventory management and the creation of appropriate inventory control systems for all organizations has become a major challenge, and this highlights the need for research in this area [3]. On the other hand, today, a large percentage of the total capital of organizations is inventory. In developed and developing countries, the capital held in inventories at any given time is very high, so the lack of control and inventory control system and proper assortment planning creates many problems for organizations, some of which are: all Organizations are faced with inventory-related costs such as maintenance costs, ordering, shortages, etc. The lack of a proper inventory control system can increase any of these costs. In some organizations, a shortage of inventory may cause the production process to stop, and the organization may face the problem of not delivering the product to customers on time and thus increase the cost of shortage. In some cases, the organization may face an excessive increase in inventory, which in this case also increases the cost of maintaining inventory; so in both cases, the lack of inventory control system and proper shelf space allocation will have negative effects on the profits of organizations [4].

Since a substantial part of customers' decisions is made at the point of sale (where only one-third of the purchasing is the result of previous planning [5], a retailer should understand how shelf-space decisions and better display of items in the store affect the purchasing behavior and demand for a product [6, 7]. The more shelf space the retailer allocates to a product, the more visible it will be, and consequently, the more demand it will face. This behavior shows the space elasticity of the demand [8, 9]. The retailers are now fully aware that the availability of the right products has a drastic effect on customer satisfaction, and an optimized assortment can guarantee their superior position in the market. Through the right assortment and proper display of the products, retailers can further improve their performance by directing customers to have unwanted purchases or buy items with a high margin of profit [10]. However, the retailers' shelf space is limited, especially nowadays, where the diversity of products has increased considerably. This increasing number of products, limited shelf space, narrow retail margins, and intensified competition have substantially promoted the position of the assortment and shelf-space planning [11].

An important parameter that significantly affects the modeling of the assortment problem is substitution. When customers do not find their favorite product, they may substitute it with another similar product [1, 2]. The willingness of customers to substitute a product with a similar one within its category is an important factor in AP; when there is the willingness to substitute a product, providing a great deal of inventory for that product is not so critical [12]. Another fundamental concern in AP is supplier selection. Selecting a good set of suppliers to collaborate with is crucial to the retailer's success. In this regard, understanding the expectations and purchasing behaviors of customers can be helpful. Generating a rich diversity in the range of products increases customer satisfaction but, on the other hand, leads to more operational costs. Thus, the right assortment should make a tradeoff between these two factors [13].

All of the aforementioned materials take on added importance when it comes to perishable products. A commodity is perishable if it is subject to deterioration in quality or quantity [14]. In dealing with perishable products, shelf-life plays an important role. Shelf life is the length of time an item can remain salable on a retailer's shelf and reflects its marketable life [15].

Generally, retailers consider dynamic pricing, planning, and inventory control models for perishable products.

As was said, assortment planning, space allocation, inventory management, and supplier selection are the most important decisions made by retailers. Many previous studies have addressed these decisions but not in an integrated manner. These determining parameters are interdependent, and certainly, a model that considers them simultaneously can provide a more workable and reliable solution. Accordingly, this paper develops an integrated mathematical model for assortment planning, supplier selection, space allocation, and inventory management of perishable products, in which the demand for a product depends on the amount of space allocated to it. Besides, this model accounts for substitution.

The remaining of this research is organized as follows. Section 2 reviews the literature, and Section 3 provides the statement of the problem and illustrates the modeling and research assumptions. Section 4 presents the solving algorithm, with a detailed description of the genetic operators. Section 5 provides the computational results and sensitivity analysis. Finally, Section 6 concludes the paper.

## 2. Literature review

Retail store operations have been of interest in the previous two decades [16, 17]. In a review paper by Mou, Robb, and DeHoratius, the related published works were categorized into seven groups in terms of decisions they focused on, including demand forecasting, assortment planning, and inventory management. Reviewing the works published up to 2016, they reported that only a few papers addressed more than one area, with an average of 1.1 areas for each paper. After that, however, researchers usually have addressed these decision areas jointly to provide more reliable solutions [18]. We review in this section such works.

Kök and Fisher investigated an AP problem under substitution. To determine the best assortment, they provided an algorithmic process that first specified the substitution behavior and demand for each product and then solved the AP problem using an iterative optimization heuristic [12]. In a more comprehensive model, Yücel et al. took into account demand substitution, supplier selection, and shelf space limitations. To demonstrate the performance of their proposed model, they developed three modified models, each of which ignored one of the three mentioned factors. As was expected, the results showed that ignoring each of these determining factors might result in inefficient assortments [13]. Honhon, Gaur, and Seshadri determined the optimal assortment and inventory levels for a given set of products with varying prices and costs and stochastic demand. They also considered customer preferences and modeled by defining customer types, that determines an ordered ranking of potential products [19]. Tan and Karabati investigated inventory management problem under demand substitution. They assumed that the substitution is made only once, and the demand is lost when the second-choice product is not available [20]. Boada-Collado and Martínez-de-Albéniz investigated the impact of inventory on the choices of the customers for a fashion retailer. Knowing how inventory level affects the choices can help the retailer in adjusting its inventory to have the maximum sales or profits [21].

Sainathan addressed pricing and inventory management for a perishable product with two-period shelf life: in the first period, it is a new product and in the second one, it is considered as "old". The retailer should determine the price for each period and the order quantity for the new product [22]. Piramuthu and Zhou assumed that how the products are handled in transit and during storage affects the shelf life of perishable products, and established a direct relationship between the demand of a product and its allocated space and quality. To track the quality of the products, they used the information provided by RFID technology. Regarding the

characteristics of new business trends making demand prediction more difficult, which in turn complicates AP [23], Talebian, Boland, and Savelsbergh proposed a stochastic dynamic programming model that simultaneously made assortment and pricing decisions. They investigated the effect of demand learning on retailers' profitability [24]. Azadi, Eksioglu, Eksioglu, and Palak addressed the inventory management and pricing of perishable products. Implying that good pricing can help retailers in reducing waste and increasing profitability, they proposed a two-stage stochastic optimization model for supplier selection, replenishment scheduling, and price markdown planning [25].

Hariga, Al-Ahmari, and Mohamed presented a mathematical model for AP and shelf space allocation. Considering shelf space and backroom storage constraints, they modeled this problem as a mixed-integer non-linear program. They also considered the impact of display location on the demand. The product assortment, display locations, order quantities, and allocated shelf space were the decision variables of their integrated model [26]. A. Hübner and Schaal included substitution and space-elasticity effects in their optimization model. The substitution was for products that were out-of-assortment or out-of-stock. They developed a specialized heuristic method that could efficiently provide near-optimal solutions and outperform the alternative approaches that sequentially planned the assortment and shelf space [8]. Flamand, Ghoniem, Haouari, & Maddah assumed that in addition to the attractiveness of shelf segments, the profitability of product categories, their expected demands, and their impulse purchase potential are also determinant of the obtained profit. To solve the developed model, they proposed a mixed-integer programming model [27]. Reisi, Gabriel, and Fahimnia presented a bi-level model for optimizing the shelf-allocation and pricing problems for a supply chain network consisting of two manufacturers at the top level and a common retailer at the bottom level. They provided a closed-form approximate solution to the lower-level problem to determine the retail prices and allocated spaces. Then, to maximize the manufacturers' profit, they incorporated this solution into the objective function of the top-level problem. The sensitivity analysis revealed that price and shelf space are critical in increasing the manufacturers' profit [28]. Karki, Guthrie, and Parikh addressed the tradeoffs between the benefits of an appropriate rack layout and product placement and the costs associated with floor space and restocking. They developed a model that jointly determines rack decisions and product decisions [29].

Kim & Moon presented a mixed-integer non-linear programming (MINLP) model for shelf-space allocation with product selection and replenishment decisions to maximize the retailer's profit. They considered space and cross-space elasticities and positioning effects on each product demand. They proposed tabu search and genetic algorithms to solve the problem [2].

To summarize, assortment planning, space allocation, inventory management, and supplier selection are the most important decisions retailers make, and they are closely related to each other. Demand substitution, space-elasticity demand, and product perishability make these decisions more complicated in the real world [30, 31]. To the best of our knowledge, no studies in the literature have considered all these aspects together. In this way, this paper develops an integrated mix-integer non-linear mathematical model for assortment planning, supplier selection, shelf space allocation, and inventory management. The proposed model considers space elasticity and substitution behavior of customers.

In contrast, in most studies conducted, this study considers perishable products and perishability costs for the retailer. The nonlinearity of the demand function makes the problem a mixed-integer non-linear model. GAMS BARON solver is hired to solve the proposed model in small and medium scales. An evolutionary genetic algorithm (GA) and an efficient local search vibration-damping optimization (VDO) algorithm are proposed for large-scale problems.

## 3. Problem statement and formulation

In this section, we first describe the problem in detail, and then, propose a methodical optimization model to solve it.

### 3.1. Problem statement

Assortment planning is implied to the set of decisions for products carried in each store at each point in time. The target of assortment planning optimization is to determine an assortment that maximizes sales or gross subject to various constraints, such as a limited budget for purchase of products, limited shelf space for displaying products, and a variety of multiple constraints such as a desire to have at least two vendors for each type of product.

Shelf-space mathematical models optimize the number of facings for items with space-elastic demand to be allocated to limited shelf space. Respective approaches aid retailers in dealing with the trade-off between more shelf space (and thus demand enhancement due to a higher number of facings) for specific items and less available space (and therefore demand decreases due to a lower number of facings) for other products.

Multi-item inventory problems are also highly relevant to the assortment planning problem. The inventory management of multiple products under shelf space limitations or budget constraints can be a critical issue that needs consideration.

Assortment, shelf-space allocation, inventory management, and supplier selection are among the most important decisions in retailing. While each one of these decisions affects the optimal value of the others, previous models have not addressed them integratedly. Accordingly, we develop a model that simultaneously addresses these factors. This model also considers the effect of inventory and display on demand for the products. The problem can be described as follows:

We have a retailer selling the products through a physical channel. The customers assess the product and buy it if it is available in the store, or they leave the store without buying if it is unavailable or undesirable. The demand for a product is a function of its inventory and the space in which it is displayed. The retailer is also involved in selecting the appropriate suppliers who can provide the products with desirable requirements. The problem deals with perishable products with a limited and fixed lifetime; when they reach the end of their life, they perish, and the retailer incurs the perishability cost. Considering these, we formulate a mathematical optimization model aiming at maximizing the sales and the retailer's profit.

This model considers these assumptions:

- The customers' demand for a product depends on the space considered for its display.

- In case of not finding the desired product, a part of the customers substitutes another product.

- The shelf space for the product assortment and display is limited.

- The costs of supplier selection, fixed cost of ordering, assortment, purchase, transport, substitution, perishability, and not satisfying the demand are included.

- The products are perishable with a limited and fixed lifetime beyond which the retailer incurs the perishability cost.

### 3.2. Proposed mathematical model

In this section, mathematical modeling along with its assumptions is described.

**Sets**:

| symbol | Definition |
|--------|------------|
| $i$ | Index of the product number |
| $j$ | Index of supplier number |
| $k$ | Index of the substituted product number |
| $t$ | Index of time |
| $g$ | Index of product lifetime |
| $N$ | Set of available products, $N = \{1, 2, \ldots, i, \ldots k, \ldots, I\}$ |
| $M$ | Set of suppliers, $M = \{1, 2, \ldots, j, \ldots, J\}$ |
| $G_i$ | Set of lifetimes of product i, $G = \{1, 2, \ldots, g, \ldots, G_i\}$ |
| $T$ | Planning horizon |

**Parameters**:

| symbol | Definition |
|--------|------------|
| $W_{ik}$ | Percentage of customers who choose product k if product i is not available |
| $C_i$ | Unit cost of buying and transporting product i |
| $OC_j$ | Fixed cost of ordering from supplier j |
| $SSC_j$ | Selecting cost of supplier j |
| $dnorm_{it}$ | Normal/usual demand for product i at time t |
| $S$ | Total shelf space available |
| $a_{ij}$ | The binary parameter is 1 if product i is supplied by the supplier j, otherwise it is zero. |
| $h_i$ | Unit holding cost for product i |
| $V0_i^g$ | Initial amount of product i with lifetime g at the store |
| $br_i$ | The width of each unit of producti in the selected display |
| $\beta_i$ | spatial elasticity of product i |
| $P_i$ | Sales price of each unit of product i |
| $G_i$ | Maximum lifetime of product i |
| $s_k$ | Penalty cost of substitution for product k |
| $\alpha_i$ | Unit cost of destroying product i |
| $\delta_i$ | Maximum number of product i in each allocated display |
| $lc_i$ | Unit cost of non − responding to the demand of product i |
| $W_{ik}$ | Percentage of customers who choose product k if product i is not available |
| $C_i$ | Unit cost of buying and transporting product i |
| $OC_j$ | Fixed cost of ordering from supplier j |

**Decision variables**:

| symbol | Definition |
|---|---|
| $y_i$ | Binary variable : it is 1 if product i is chosen; otherwise, it is 0. |
| $yy_{ti}$ | Binary variable : it is 1 if product i is present in period t; otherwise, it is 0. |
| $o_{jt}$ | Binary variable : it is 1 if supplier j is ordered at time t; otherwise, it is 0. |
| $z_j$ | Binary variable : it is 1 if supplier j is chosen; otherwise, it is 0. |
| $x_i^t$ | Required amount of product i at time t |
| $m_{ikt}^g$ | Amount of product i with lifetime g allocated to the demand of product k at time t |
| $V_{it}^g$ | Amount of inventory of product i with lifetime g at the end of period t |
| $f_{it}$ | Number of displays that can be allocated to product i at time t |
| $d_i^t$ | Total demand for product i at time t |
| $q_{it}$ | Unsatisfied demand for product i at time t |

$$MAX\ TP = TR - TCO - TCSS - TCP - TCI - TCS - TCU - TCLS \tag{1}$$

$$TR = \sum_{i \in I} \sum_{k \in K} \sum_{g \in G_i} \sum_{t \in T} p_i \cdot (m_{ikt}^g) \tag{2}$$

$$TCO = \sum_{j \in J} \sum_{t \in T} oc_j \cdot o_{jt} \tag{3}$$

$$TCSS = \sum_{j \in J} ssc_j \cdot z_j \tag{4}$$

$$TCP = \sum_{i \in I} \sum_{t \in T} c_i \cdot x_{it} \tag{5}$$

$$TCI = \sum_{i \in I} \sum_{g \in G_i} \sum_{t \in T} \frac{V_{it}^g}{2} \cdot h_i \tag{6}$$

$$TCS = \sum_{i \neq k \in I} \sum_{k \in K} \sum_{g \in G_i} \sum_{t \in T} m_{ikt}^g \cdot s_k \tag{7}$$

$$TCU = \sum_{i \in I} \sum_{t \in T} (c_i + \alpha_i) V_{it}^G \tag{8}$$

$$TCLS = \sum_{i \in I} \sum_{t \in T} lc_i \cdot q_{it} \tag{9}$$

*s.t*:

$$z_j \geq a_{ij}.y_i \quad \forall j, i \tag{10}$$

$$V_{it}^g = V_{i(t-1)}^{g-1} - \sum_{k \in K} ms_{ikt}^g \quad \forall i, \forall g \in G_i, g \geq 2, \forall t \tag{11}$$

$$V_{it}^g = x_{it} - \sum_{k \in K} ms_{ikt}^g \quad \forall i, \forall g \in G_i, g = 1, \forall t \tag{12}$$

$$V_{it}^g = V0_i^g \quad \forall i, \forall g, t = 0 \tag{13}$$

$$\sum_{i \in I} \sum_{g \in G_i} (m_{ikt}^g) + q_{kt} = d_{kt} \quad \forall k, \forall t \tag{14}$$

$$\sum_{g \in G_i} m_{ikt}^g \leq (d_{kt} - \sum_{g \in G_i} \sum_{i=k \in I} m_{ikt}^g) w_{ik} \quad \forall i, \forall k \neq i, \forall t \tag{15}$$

$$\sum_{i \in I} f_{it}.br_i \leq S \quad \forall t \tag{16}$$

$$d_{it} = dnorm_{it}(1 - yy_{it}) + dnorm_{it}(f_{it}.br_i)^\beta \quad \forall i, t \tag{17}$$

$$\sum_{g \in G_i} \frac{(V_{it}^g + x_{it} + V_{i(t-1)}^g)}{2\delta_i} = f_{it} \quad \forall i, t \tag{18}$$

$$\sum_{t \in T} o_{jt} \leq M.z_j \quad \forall j \tag{19}$$

$$\sum_{t \in T} x_{it} \leq M.y_i \quad \forall i \tag{20}$$

$$\sum_{t \in T} x_{it} \geq y_i \quad \forall i \tag{21}$$

$$x_{it}.a_{ij} \leq M.o_{jt} \quad \forall i, j, t \tag{22}$$

$$f_{it} \leq Myy_{it} \quad \forall i, t \tag{23}$$

$$f_{it} \geq yy_{it} \quad \forall i, t \tag{24}$$

$$m_{ikt}^g, v_{it}^g, f_{it}, d_{it}, x_{it} \geq 0 \quad \forall i, g, k, m, t \tag{25}$$

$$o_{jt}, z_j, y_i, yy_{it} = \{0, 1\} \quad \forall i, t \tag{26}$$

The objective function in Eq (1) represents the profit of operations during the period under study. Eq (2) calculate sales income, it should be noted in cases i = k it consider the direct sale of product i and in cases i≠k consider the sale of product i that is allocated to the demand of

product k, (3)–(9) show respectively the amount of fixed cost of ordering, cost of supplier selection, purchase costs, average inventory holding costs, penalty cost for substituting another product instead of the desired product, cost incurred due to the expired date and the corruption of the product, and penalty cost of not responding to the demands.

Where supplier selection costs explain contract registration costs and ordering costs show supplying products costs from a specific supplier. Another hand, the penalty cost for substituting is a cost based on creating customer distrust. The penalty cost of not responding to the demands represents demand loss cost.

Constraint (10) refers to the allocation of products to the supplier (displaying which suppliers supply what products). Constraints (11) to (13) show the inventory capacity and its transfer to subsequent periods. Constraint (14) ensures that the demand can be satisfied as much as the inventory (there is no possibility of allocating more than the inventory to the demand).

Constraint (15) shows the substitution constraint for the desired product: if product k is not selected, its demand is responded by another product according to the substitution matrix at the first level. This constraint ensures that the amount of product $i$ used for satisfying the demand for k (based on the substitution matrix) is less than the unsatisfied demand for k (inventory of k in each period is subtracted from its demand).

Constraint (16) is the shelf space constraint. Constraint (17) is the dependence of the product demand on the display allocated to that product. Constraint (18) calculates the average amount of product displayed in each period. Constraints (19)–(22) ensure the product selection and supplier selection if it's ordered. Constraint (23) and (24) ensure the product have a face if it's presented in each period. Constraints (25) and (26) also specify the type of decision variables.

## 4. Metaheuristic solution approaches

Due to including large-scale binary programming, assortment planning (AP) and shelf space allocation (SSA) is an NP-hard problem [1, 27, 32]. Therefore, it is obvious that the integrated AP-SSA problem with the inventory control problem of this study is an NP-hard problem. Therefore, to solve the problem in large-scale instances, we propose two metaheuristic solvers, one of which is a population-based genetic algorithm (GA) and the other is single-based local search vibration-damping optimization (VDO).

To apply the proposed GA and VDO methods, in the following, we firstly explain solution encoding and decoding strategy, initial solution generating mechanism, and neighborhood search operators, and then, we provide the flowchart of the metaheuristic solution method.

### 4.1. Solution encoding/decoding and fitness

For solution representation, we consider a seven-part structure including 1) a vector of size N for product selection, 2) a vector of size M for supplier selection, 3) a matrix of size N*T for the ordering period of each product, 4) a matrix of size N*T for the ordering amount of each product in the order period, 5) a matrix of size N*M for the fraction of each product order allocated to the suppliers, 6) a vector of size N for the space allocated to each selected product, and finally, 7) a matrix of size $N^*G^*T$ for the amount of product supply/sales in each lifetime to satisfy demand in each period. It is necessary to explain that variables such as shortage, inventory, and substitution rate are functions of the above decisions and product demand.

Part 1) A vector (named P1) of size N for product selection. This vector includes continuous real numbers between 0 and 1. The numbers are rounded, and then the products corresponding to the values 1 on vector cells are selected (See part 1 of Fig 1).

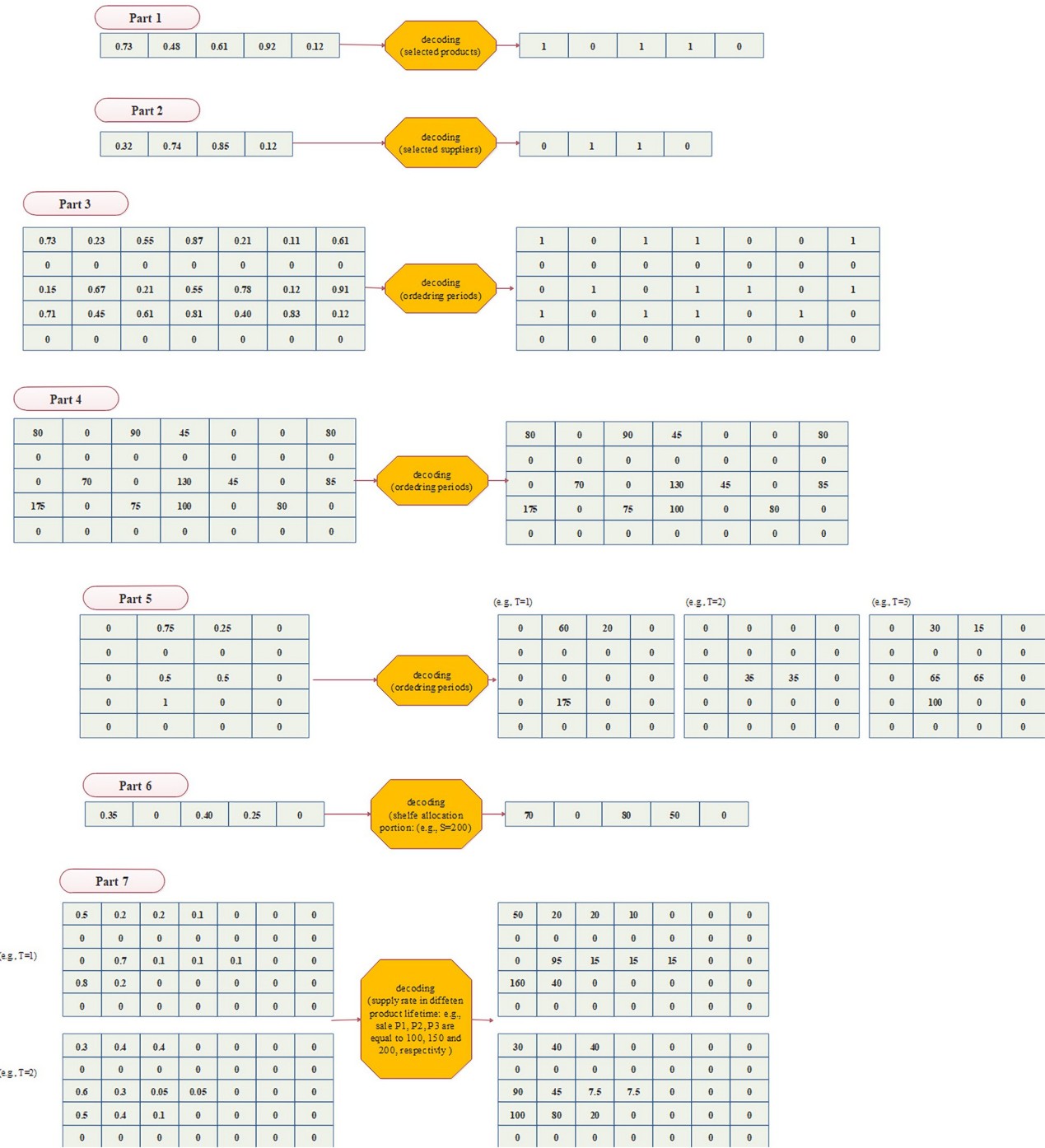

**Fig 1. The proposed solution encoding and decoding to apply the metaheuristic search.**

Part 2) A vector (named P2) of size M for supplier selection. This vector includes continuous real numbers between 0 and 1. The numbers are rounded, and then the suppliers corresponding to the values 1 on vector cells are selected (See part 2 of Fig 1).

Part 3) A matrix (named P3) of size N*T for the product ordering periods. This matrix includes continuous real numbers between 0 and 1. For each raw (product), the numbers

are rounded and then, the column (period) with values 1 are considered as ordering periods (See part 3 of Fig 1). Note that if a cell of the vector P1 is 0, the corresponding value in rows of this matrix is also 0.

Part 4) A matrix (named P4) of size N*T for the products' economic order at the ordering periods. This matrix includes positive real numbers limited by shelf space capacity (See part 4 of Fig 1). Note that if a cell of matrix P3 is 0, the corresponding value in matrix P4 is also 0.

Part 5) A matrix (named P5) of size N*M for order allocation to the selected suppliers. This matrix includes continuous real numbers between 0 and 1, where the sum of values in each row of this matrix is equal to 1 (See part 5 of Fig 1). Note that if a cell of matrix P1 or P3 is 0, the corresponding row and column in matrix P5 are also 0.

Part 6) A vector (named P6) of size N for the space allocated to each selected product. This vector includes continuous real numbers between 0 and 1, where the sum of vector cells is equal to 1 (See part 6 of Fig 1). Note that if a cell of vector P1 is 0, the corresponding value in vector P6 is also 0.

Part 7) A matrix of size N*G*T for determining the amount of product supply/sale rate in each lifetime. Each row of this matrix includes continuous real numbers between 0 and 1, where their sum is equal to 1. In this matrix, we define $G$ as the maximum of $G_i$ concerning all products or $G = T$ (See part 7 of Fig 1). Note that if a cell of vector P1 is 0, the corresponding value in this matrix is also 0. It is obvious that the sale of a product with a life bigger than $G_i$ is equal to 0 because the product deteriorates.

After encoding (See Fig 2), we first determine the value of decision variables such as product selection, supplier selection and order allocation, shelf space allocation, ordering period and quantity, etc., and then with the consideration of active demand and product availability, the dependent variables such as product substitution are determined to reduce shortage variables. Finally, the inventory level of all products without demand and the objective function value are determined. In other words, fitness evaluation can be calculated by the following procedure:

*Step 1)* **Do encoding**.

*Step 2)* **Find the value of the decision-making variables as** Fig 2.

*Step 3)* **Calculate the objective function value, temporally**.

*Step 4)* **Check the constraint satisfaction**.
> **If it is possible, reduce their violation and shortage using the dependent variables such as substitution and product inventory**.
> **Else, add shortage cost to the objective function**.

*Step 5)* **Calculate the inverse value of the final objective function as the fitness of the solution (Note that the objective function is cost minimization)**.

## 4.2. Initial solution generation

To generate the initial solution, the cells of matrixes and vectors P1, P2, . . ., P7 are filled randomly in their authorized ranges. Note that we repair some vectors or matrixes if some

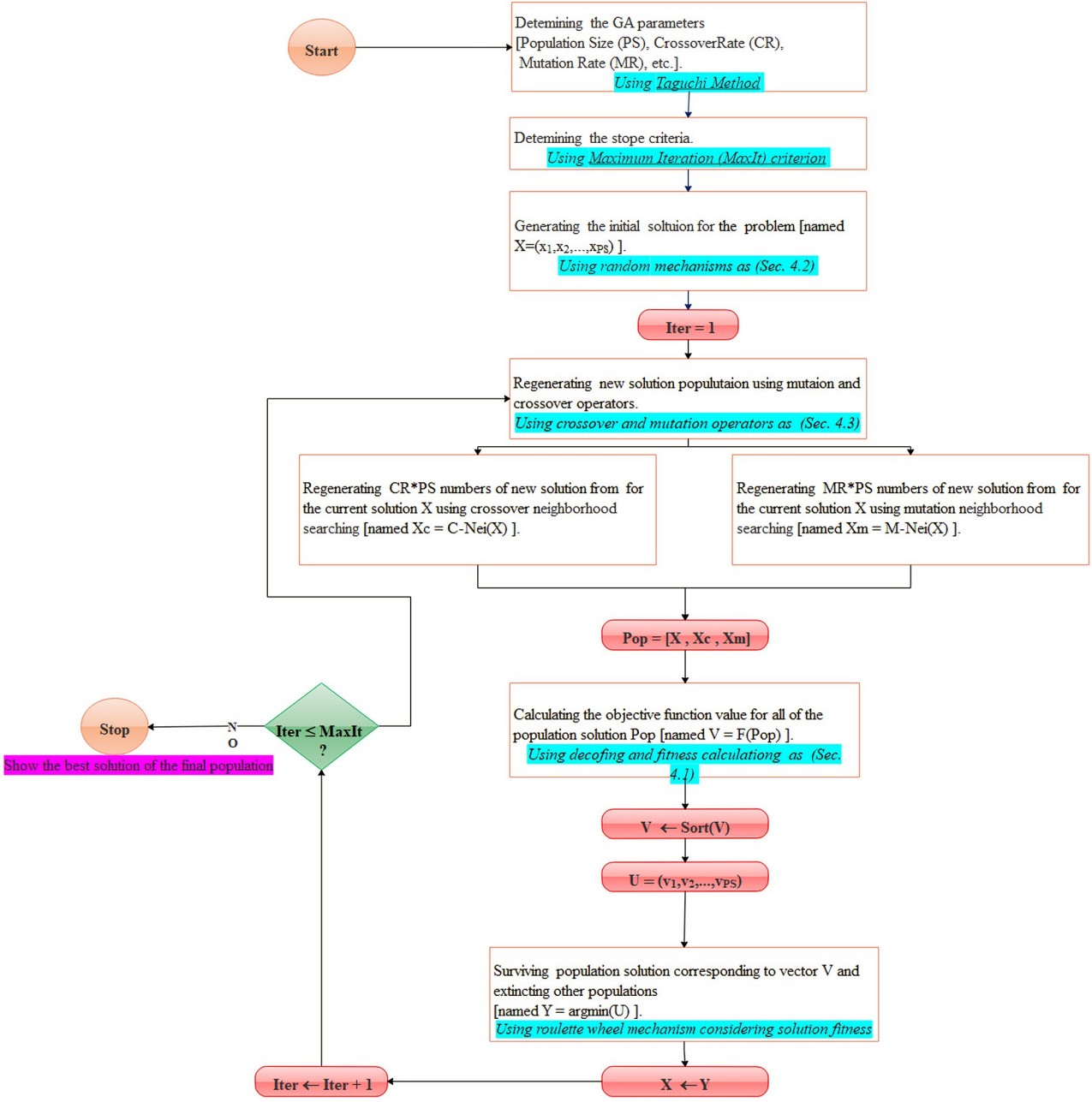

**Fig 2. The flowchart of the proposed GA solver in solving the problem.**

constraint be violated. For example, the summation of the cells in vector P6 should be equal 1, and if a cell of vector P1 is 0, the corresponding value in this vector should also be 0.

## 4.3. Neighborhood searching

Since we used continuous representation in the proposed encoding, we apply linear convex neighborhood search operators $\otimes$ and $\circledast$ as crossover (globally) and mutation (locally) search

strategies. For each part of the seven-part solution structure, we define these operators as follow:

$$
\begin{cases}
P^1(i) \otimes P^2(i) = vP^1(i) + (1-v)P^2(i) \mid \text{v} = \text{random}(0,1); \text{ i} = 1,2,\ldots,7 \\
\quad\quad \circledast P^1(i) = (1+\sigma)P^1(i) \mid \sigma = \text{random}(-0.1,0.1); \text{ i} = 1,2,\ldots,7 \\
P^1(i) \otimes P^2(i) = (1-v)P^1(i) + v\,P^2(i) \mid \text{v} = \text{random}(0,1); \text{ i} = 1,2,\ldots,7 \\
\quad\quad \circledast P^2(i) = (1+\sigma)P^2(i) \mid \sigma = \text{random}(-0.1,0.1); \text{ i} = 1,2,\ldots,7
\end{cases}
$$

in which, $P^1(i)$ and $P^2(i)$ are the i[th] part of two different solution structures.

### 4.4. GA mechanism and flowchart

Genetic Algorithms (GA)s refer to a family of computational models inspired by evolution that encode a potential solution to a simple chromosome-like data structure and apply recombination/reproduction operators, named crossover and mutation, to these structures to preserve critical information. In the management or engineering optimization [33, 34]. GA is the most well-known metaheuristic function optimizer that has been used in various fields such as selection, scheduling, etc. In general, this method is one of the most efficient ways to solve optimization problems with pure or mixed-binary programming formulation. Population Size (PS), Crossover Rate (CR), and Mutation Rate (MR) are some of the main affecting factors of GA. Fig 2 illustrates the flowchart of the proposed GA mechanism to search the solution space of the research problem.

### 4.5. VDO mechanism and flowchart

In physics, vibration can be defined as the repetitive motion of an object around an equilibrium position. Vibration damping is a reduction process of the amplitude of oscillation, tending to zero over time [35]. There is a useful relation between the vibration damping process and optimization solvers. In the solving methodologies area, Mehdizadeh, Tavakkoli-Moghaddam and Yazdani firstly developed a new metaheuristic algorithm, namely Vibration Damping Optimization/Optimizer (VDO). VDO is a local search iterative method inspired by the Simulated Annealing (SA) algorithm and is created based on the concept of the vibration damping phenomena. To optimize an optimization problem, the VDO method is affected by four main factors consisting of initial amplitude (A), the number of iteration at each amplitude (N), damping coefficient ($\gamma$), and vibration standard deviation ($\sigma$). These factors should be tuned by a systematic method as Taguchi to improve searching performance. Fig 3 illustrates the flowchart of the proposed VDO mechanism to find the best problem solution [36].

## 5. Numerical result and computational analysis

The proposed mathematical model has a profit objective function for assortment planning, shelf space allocation, and inventory management of perishable products. It is solved with the GAMS 24.1.2 (BARON Solver) and Matlab 2019(b) software using a core i5 CPU (2GHz frequency), 8 GB RAM processor. Table 1 displays the parameters of a small-sized problem.

### 5.1. A small-sized instance

In this section, we solve a small-scale problem. The example we consider includes four different products along with four substituting products, two suppliers, and four periods. The life cycle of the product is five periods.

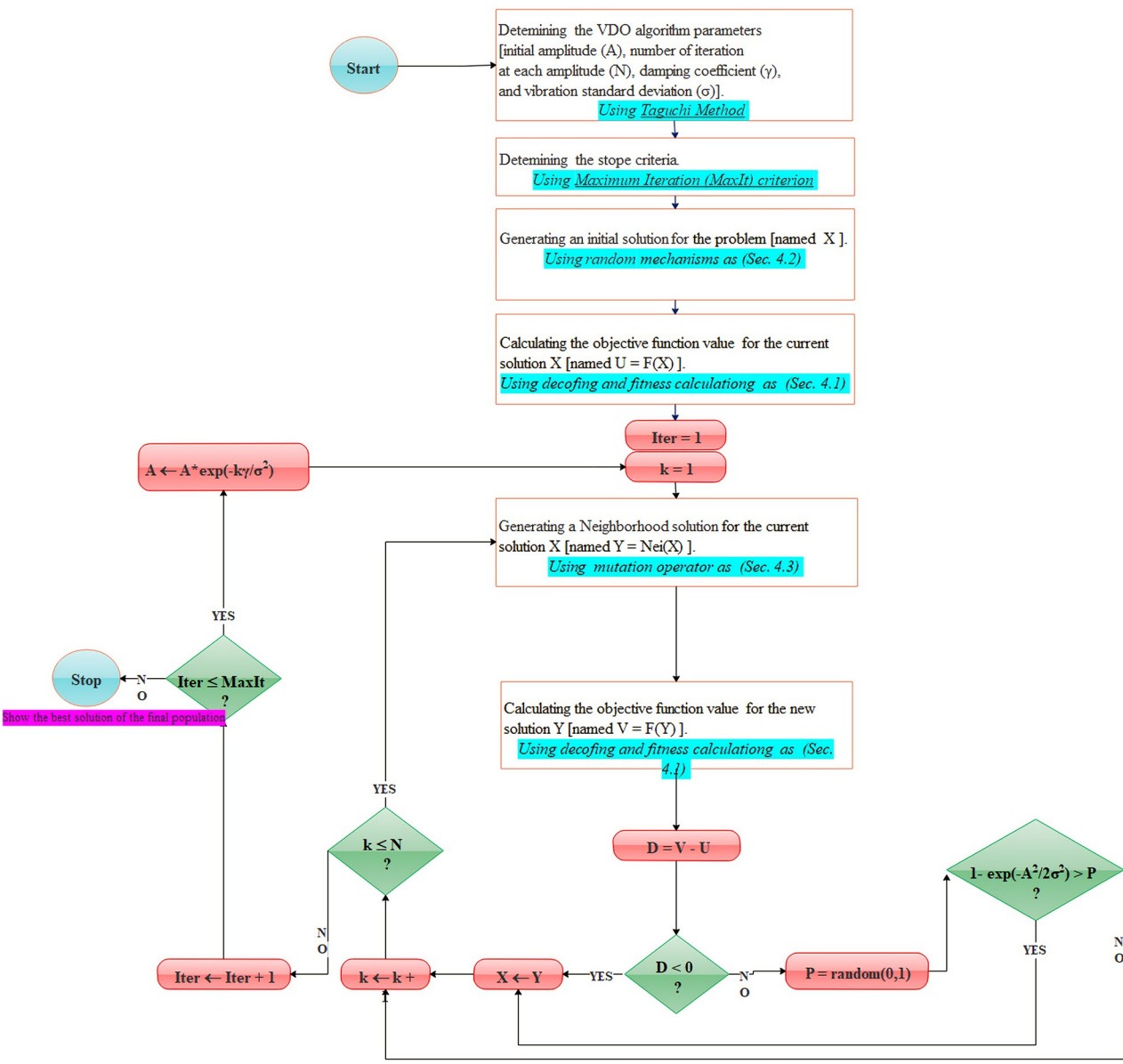

**Fig 3. The flowchart of the proposed VDO solver in solving the problem.**

The model of this sample problem was solved using GAMS BARON Solver, whose results including different cost values and the total profit are given in Fig 4.

According to Fig 4, the total profit is 101219.688 obtained from total sales as large as $184477.51. The total cost of this problem is derived by fixed ordering cost $39, supplier selection cost $18, purchasing cost $77727.69, holding cost $273.51, penalty cost for the substitution due to not selecting the product $3512.98, cost of product expiration and destroying $466.83, and finally cost of not responding the demand $1219.52. In this problem, two potential suppliers were selected, for which the ordering scheme has been shown for different periods in Table 2.

**Table 1. The parameters of sample problem.**

| $dnorm_i^t$ | t = 1 | t = 2 | t = 3 | t = 4 | | $w_{ik}$ | k = 1 | k = 2 | k = 3 | k = 4 |
|---|---|---|---|---|---|---|---|---|---|---|
| i = 1 | 60 | 40 | 30 | 6 | | i = 1 | 0 | 0.269 | 0.700 | 0.160 |
| i = 2 | 90 | 20 | 45 | 23 | | i = 2 | 0.170 | 0 | 0.113 | 0.200 |
| i = 3 | 30 | 50 | 15 | 10 | | i = 3 | 1 | 0.252 | 0 | 0.228 |
| i = 4 | 25 | 60 | 15 | 9 | | i = 4 | 0.234 | 0.187 | 0.172 | 0 |
| | $c_i$ | $\delta_i$ | $\alpha_i$ | $h_i$ | $lc_i$ | $br_i$ | $\beta_i$ | $p_i$ | $G_i$ | $s_i$ |
| i = 1 | 10 | 10 | 1 | 1 | 60 | 8 | 0.2 | 50 | 2 | 30 |
| i = 2 | 90 | 20 | 2 | 2 | 110 | 15 | 0.1 | 180 | 5 | 28 |
| i = 3 | 60 | 5 | 2 | 3 | 100 | 26 | 0.3 | 140 | 3 | 17 |
| i = 4 | 80 | 4 | 2 | 2 | 50 | 12 | 0.1 | 110 | 5 | 25 |
| | $oc_j$ | $ssc_j$ | | | | $a_{ji}$ | k = 1 | k = 2 | k = 3 | k = 4 |
| j = 1 | 5 | 10 | | | | j = 1 | 0 | 1 | 0 | 1 |
| j = 2 | 6 | 8 | | | | j = 2 | 1 | 0 | 1 | 0 |

From this table, it is found that only in period 2 and for supplier 2 there is no order. The demands for the products at each period are shown in Table 3. Then, the total demands for different products and their available displays at each period are shown in Table 4.

According to this table, the total demands are different from the usual demands for the products. This is due to the spatial elasticity of the products and the width of each product at the allocated display.

## 5.2. Some parameter sensitivity analysis

To investigate the effect of changing the parameters on the output variables and profit function, we conduct the sensitivity analysis (SA) of the problem under the changes of the demand, the purchasing and transportation costs, the spatial elasticity of the product, the maximum

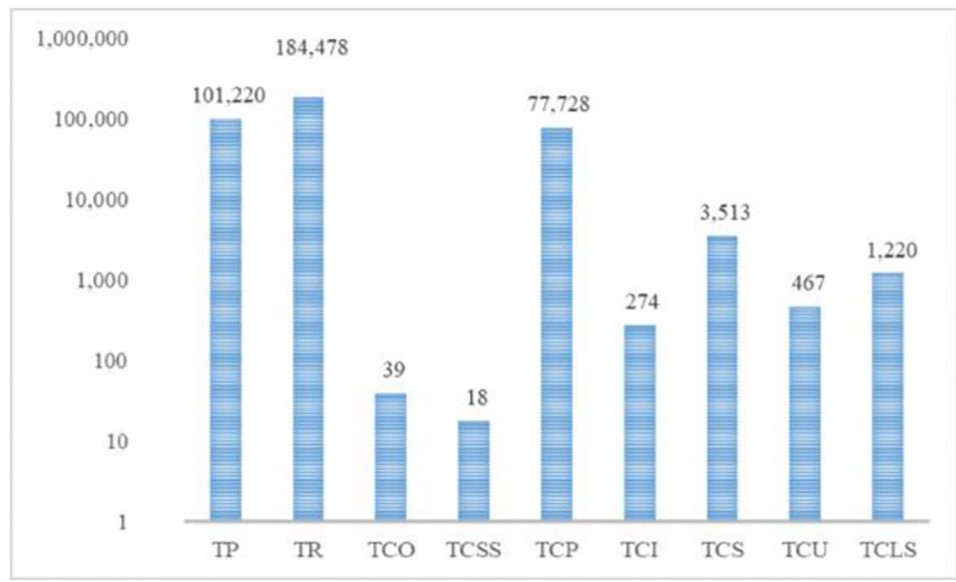

**Fig 4. The different cost values and total profit of the small-scale sample problem.**

**Table 2. The ordering for the suppliers at different periods.**

| $o_{jt}$ | t = 1 | t = 2 | t = 3 | t = 4 |
|---|---|---|---|---|
| j = 1 | 1 | 0 | 1 | 1 |
| j = 2 | 1 | 1 | 1 | 1 |

**Table 3. The demand quantity for each product at each period.**

| $X_{it}$ | t = 1 | t = 2 | t = 3 | t = 4 |
|---|---|---|---|---|
| i = 1 | 214 | 37 | 42 | 0 |
| i = 2 | 159 | 0 | 90 | 8 |
| i = 3 | 273 | 343 | 173 | 20 |
| i = 4 | 5 | 0 | 29 | 5 |

number of products at each display, the total capacity of the shelf, and price. In these analyses, we changed the values of these parameters by 10, 30, and 50 percent from their nominal values (Table 1). The results of these changes are given in their corresponding tables.

**5.2.1 Demand SA.** Table 5 displays the changes made in costs and sales profits for the changes in the demand parameter. The behavior of these changes can be seen well in Fig 5.

According to Table 5 and Fig 5, the sales income and profit increased by the increase in demand. By the demand increase, the amount of inventories at the end of periods reduced and thus the purchasing costs increased. On the other hand, due to the limited space of the shelves to be allocated to the products, the shortage and its associated costs increased.

**5.2.2 Price SA.** Table 6 displays the changes in the costs and sales profits due to the changes in the price parameter. The behaviors of these changes are shown in Fig 6.

**Table 4. The total demands and their available displays at each period.**

| | $f_{it}$ | | | | $d_{it}$ | | | |
|---|---|---|---|---|---|---|---|---|
| | t = 1 | t = 2 | t = 3 | t = 4 | t = 1 | t = 2 | t = 3 | t = 4 |
| i = 1 | 12 | 3 | 3 | 2 | 150 | 77 | 57 | 11 |
| i = 2 | 5 | 1 | 3 | 1 | 138 | 25 | 65 | 28 |
| i = 3 | 31 | 37 | 19 | 3 | 223 | 392 | 96 | 38 |
| i = 4 | 20 | 1 | 4 | 1 | 35 | 67 | 22 | 12 |

**Table 5. The costs and sales profits for the changes in demand.**

| $dnorm_i^t$ | -50% | -30% | -10% | 0 | 10% | 30% | 50% |
|---|---|---|---|---|---|---|---|
| TP | 50107.4 | 72000.35 | 92923.72 | 101219.6 | 108488.8 | 122676.5 | 135254.32 |
| TR | 98891.9 | 142911.2 | 176387.12 | 184477.5 | 201180.0 | 233681.9 | 263001.46 |
| TCO | 28 | 39 | 44 | 39 | 44 | 44 | 44 |
| TCSS | 18 | 18 | 18 | 18 | 18 | 18 | 18 |
| TCP | 42740.92 | 63333.19 | 77931.34 | 77727.69 | 84867.36 | 99481.24 | 111693.39 |
| TCI | 574.13 | 405.35 | 245.03 | 273.51 | 277.58 | 225.78 | 193.39 |
| TCS | 4569.59 | 6245.98 | 4776.19 | 3512.98 | 5202.16 | 8211.81 | 11735.001 |
| TCU | 853.91 | 869.36 | 448.82 | 466.83 | 526.995 | 650.48 | 762.837 |
| TCLS | 0 | 0 | 0 | 1219.52 | 1755.15 | 2374.08 | 3300.517 |

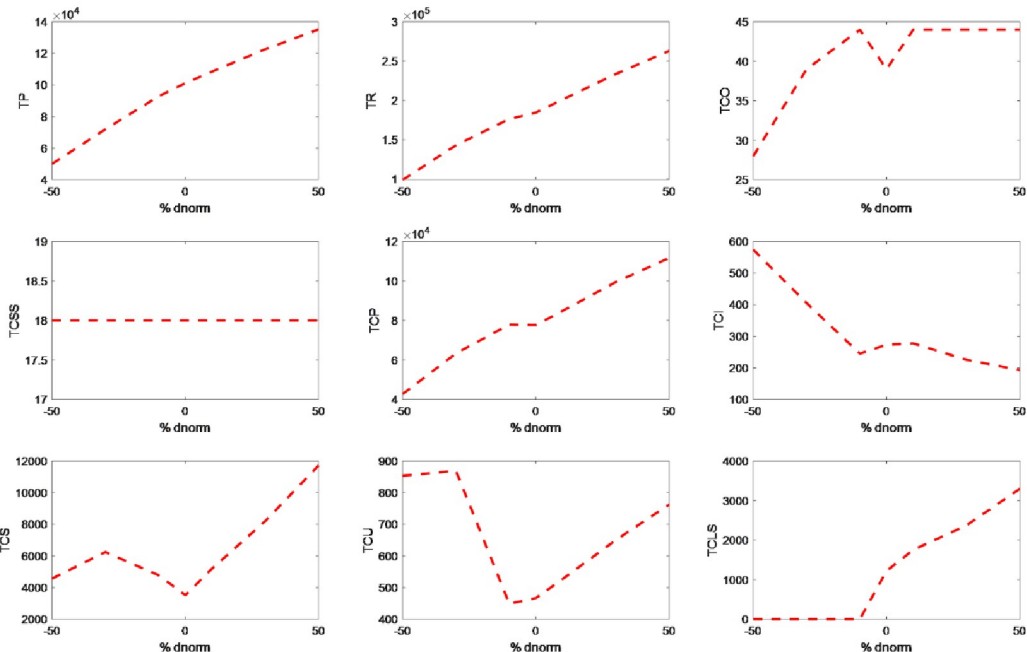

**Fig 5. The costs and sales profits behavior for the changes in the demand parameter.**

According to Table 6 and Fig 6, by the increase in the prices, the sales income and consequently the profit of the whole network increased. Besides, due to the price increase, the final inventories and their associated costs increased. On the other hand, due to the limited space of the shelves to be allocated to the products, the shortage and its associated costs increased. This is while due to the increased income from the sales, the purchasing price and penalty cost for not selecting the product has increased. Thus, the customers in this system look for substituting products.

**5.2.3 Shelf space capacity SA.** Table 7 displays the changes in the costs and sales profits due to the changes in the total available shelf space. Fig 7 shows the behaviors of these changes well.

It is observed that by the increase of the shelf space, the amount of storable products has increased, increasing consequently the holding cost. Moreover, with the increase of the products on the shelves, the shortage cost reduces. On the other hand, the order amount of the

**Table 6. The changes in the costs and sales profits due to the changes in the price.**

| $p_i$ | -50% | -30% | -10% | 0 | 10% | 30% | 50% |
|---|---|---|---|---|---|---|---|
| TP | 13012.8 | 47615.72 | 82932.88 | 101219.6 | 120145.4 | 159277.4 | 199134.21 |
| TR | 85997.7 | 121918.08 | 163894.0 | 184477.51 | 212831.7 | 257358.18 | 300451.756 |
| TCO | 38 | 39 | 39 | 39 | 44 | 38 | 38 |
| TCSS | 18 | 18 | 18 | 18 | 18 | 18 | 18 |
| TCP | 70313.8 | 71928.38 | 76362.24 | 77727.69 | 83460.68 | 87694.75 | 90584.413 |
| TCI | 198.97 | 207.49 | 250.38 | 273.51 | 275.11 | 360.48 | 426.42 |
| TCS | 1274.3 | 1116.02 | 2976.009 | 3512.98 | 6920.00 | 7750.28 | 7990.646 |
| TCU | 0 | 0 | 322.049 | 466.83 | 596.89 | 879.33 | 1171.401 |
| TCLS | 1141.78 | 993.45 | 993.454 | 1219.52 | 1371.51 | 1339.86 | 1088.661 |

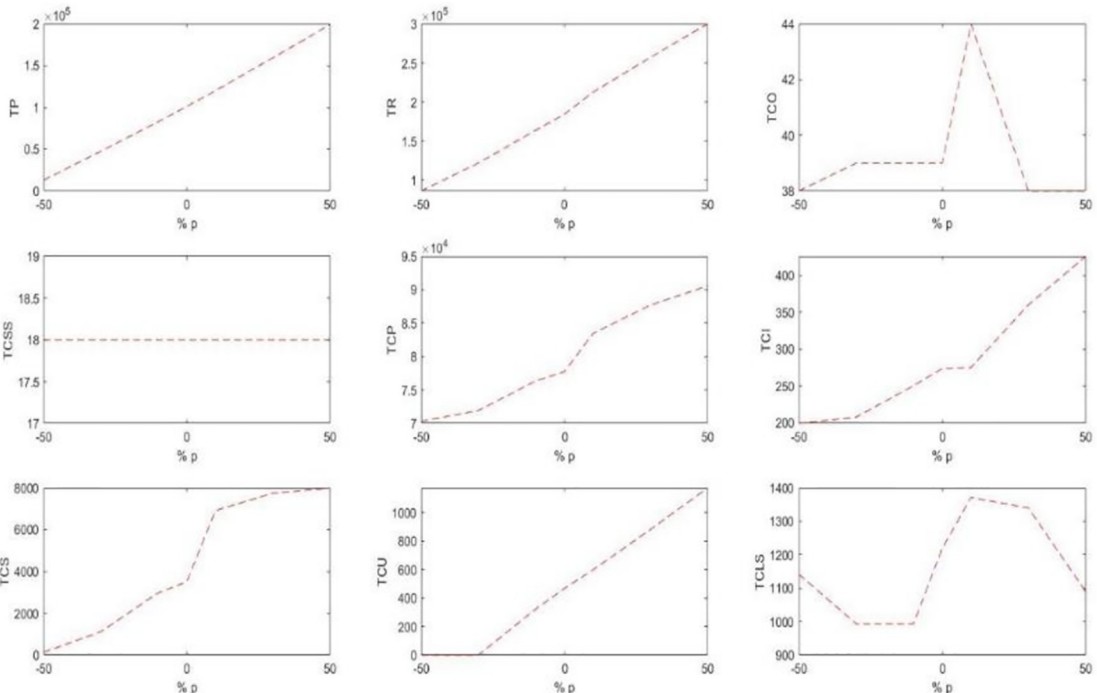

**Fig 6. The trend of changes in the costs and sales profits due to the changes in the price.**

products to be stocked on the shelves increases because of the increased space. More goods on the shelves with a fixed cost increase the sales and therefore the income and profit.

**5.2.4 Lifetime SA.** Fig 8 shows the behaviors of the changes in the sales profits due to the changes in the current lifetime products. Generally, It is observed that the amount of sales profits has increased by the increase of the lifetime. Sales profit decreases with a steeper slope for reducing product lifetime due to growing perishable costs. Sales profits have increased by the rise in the lifetime, but it has a lower pitch because the inventory costs are growing.

## 5.3. Evaluation of the proposed GA and VDO metaheuristic solvers

In the first step in evaluating the proposed GA and VDO solution approaches, the case of the DRCFJSS problem is solved using these algorithms and their results are compared with the

**Table 7. The costs and sales profits obtained by the changes in the total shelf space parameter.**

| S | -50% | -30% | -10% | 0 | 10% | 30% | 50% |
|---|---|---|---|---|---|---|---|
| TP | 77885.81 | 88715.46 | 97295.73 | 101219.68 | 104377.0 | 108526.04 | 110923.30 |
| TR | 149837.4 | 166814.95 | 178764.4 | 184477.51 | 194669.4 | 213713.90 | 223811.28 |
| TCO | 44 | 44 | 39 | 39 | 44 | 39 | 39 |
| TCSS | 18 | 18 | 18 | 18 | 18 | 18 | 18 |
| TCP | 62102.90 | 69803.60 | 74932.84 | 77727.69 | 85007.86 | 96045.73 | 101507.47 |
| TCI | 85.009 | 164.512 | 249.449 | 273.51 | 287.303 | 391.253 | 523.355 |
| TCS | 7465.516 | 5897.446 | 4320.034 | 3512.98 | 4016.207 | 7769.028 | 9445.784 |
| TCU | 408.428 | 466.876 | 467.114 | 466.83 | 469.526 | 924.849 | 1354.367 |
| TCLS | 1827.82 | 1705.057 | 1442.25 | 1219.52 | 449.492 | 0 | 0 |

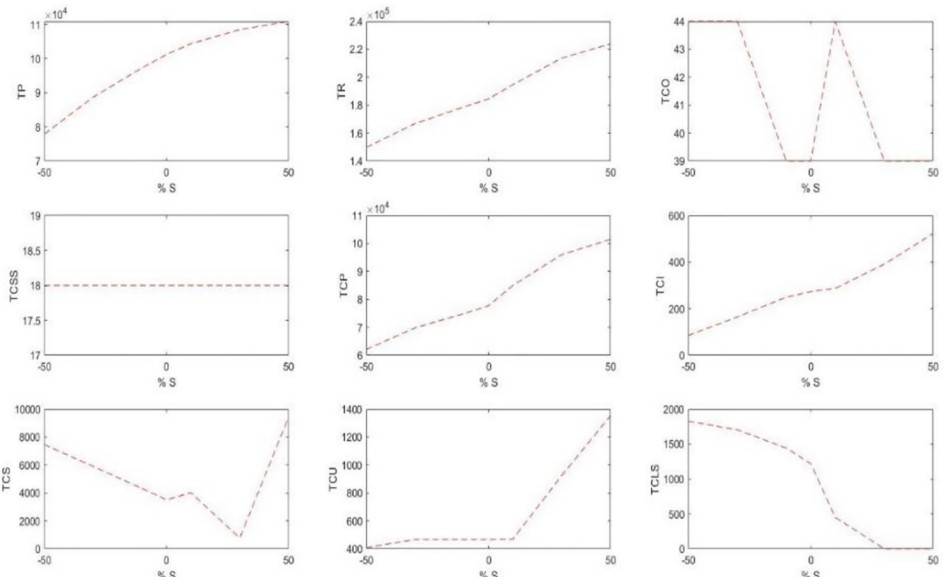

**Fig 7. The trend of changes in the costs and sales profits due to the changes in the total shelf space.**

global optimum solution obtained by GAMS. To this purpose, first, utilizing Taguchi method and sensitivity analysis, the GA effective parameters such as Population Size (PS), Crossover Rate (CR), and Mutation (MR) were tuned as [PS = 120, CR = 0.85, MR = 0.15], and the VDO parameters including initial amplitude (A), number of iteration at each amplitude (N), damping coefficient ($\gamma$), and vibration standard deviation ($\sigma$) were tuned as [A = 15, N = 100, $\gamma$ = 0.2, $\sigma$ = 2.5].

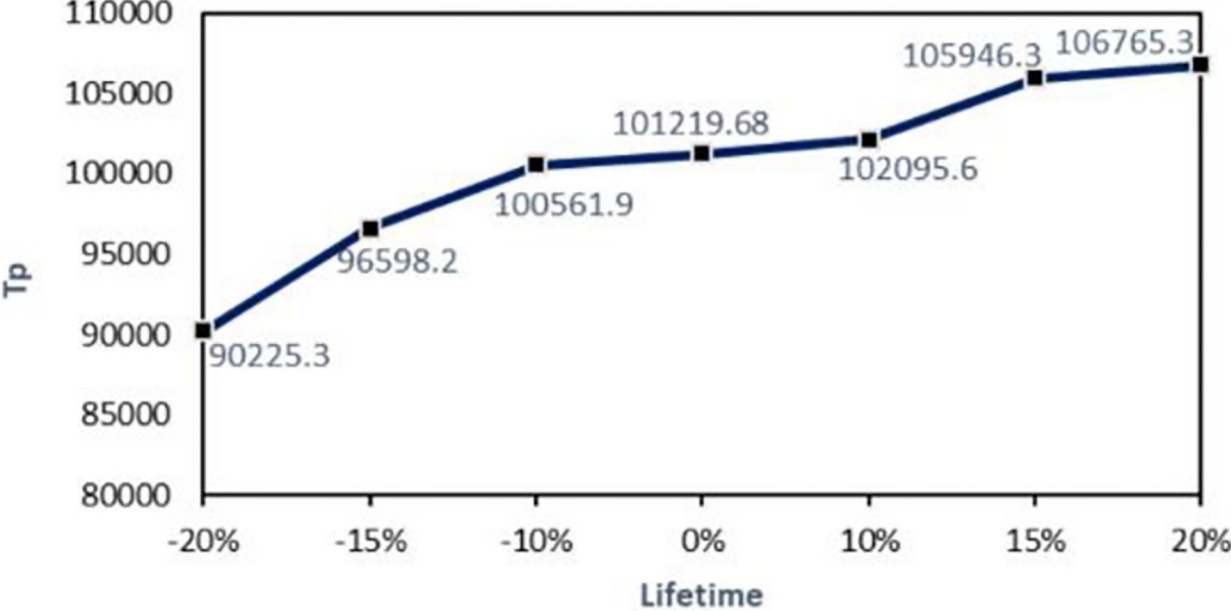

**Fig 8. The trend of changes in the sales profits due to the changes in the lifetime.**

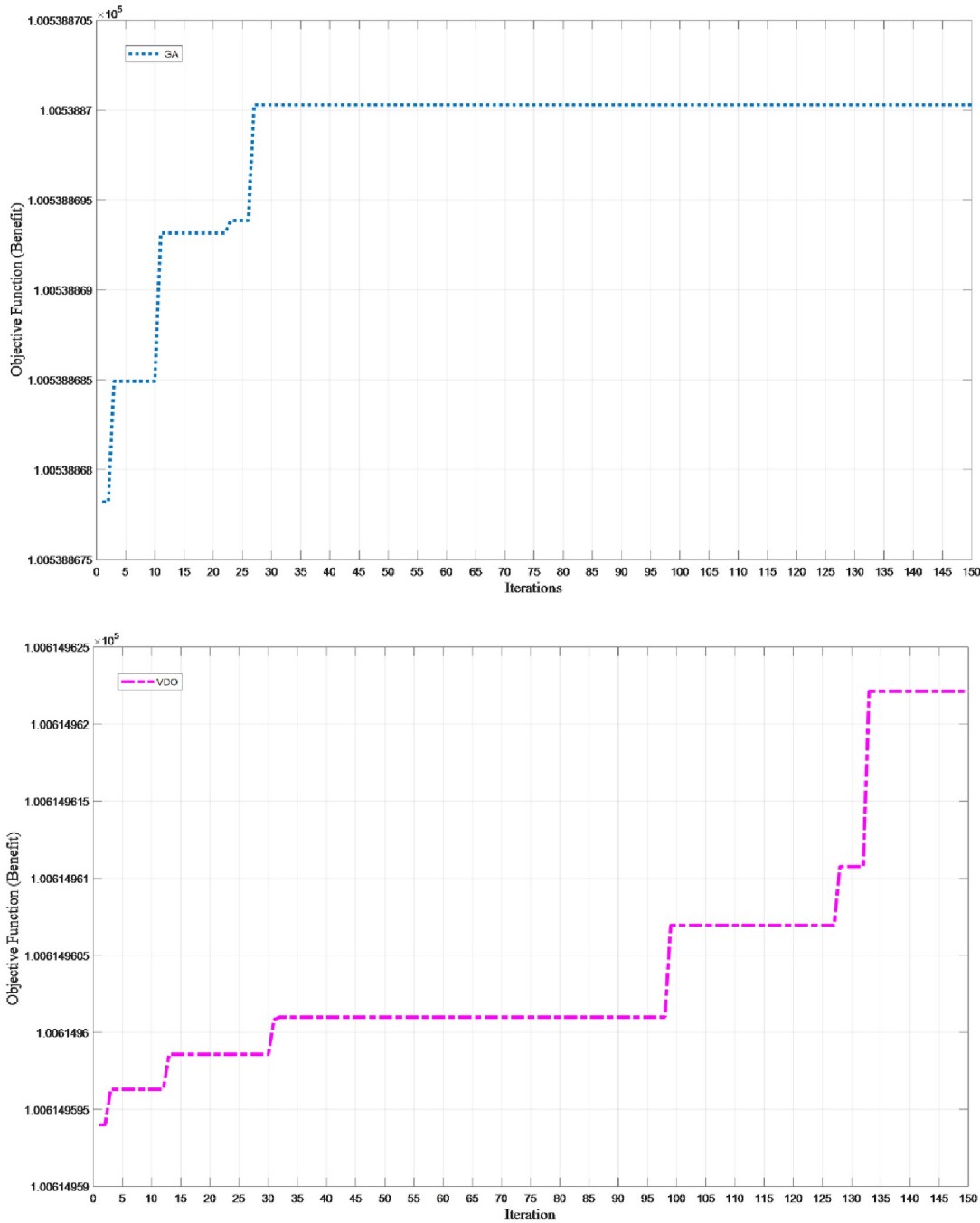

**Fig 9. The convergence of the metaheuristic algorithms in 150 consecutive iterations.**

**5.3.1 Small-sized test problem.** After tuning the parameters of the metaheuristic algorithms, we solved some small-sized test problems by the GA and VDO algorithms to explore the results of the initial chromosome and compare the relative percentage difference. Fig 9 displays the convergence of these algorithms in 150 consecutive iterations.

**Table 8. The comparison of the exact and metaheuristic algorithms for the small-sized test problems.**

| Solution Method | Objective Function | Cpu-Time (sec) | Gap (%) |
|---|---|---|---|
| GA | 100538.86 | 4.03 | 0.6726 |
| VDO | 100614.95 | 2.37 | 0.5974 |
| GAMS (Baron Solver) | 101219.68 | 763.15 | - |

**Table 9. The costs and incomes of designing the system with different solving methods.**

| Solution Method | GA | VDO | GAMS (Baron Solver) |
|---|---|---|---|
| TP | 100538.86 | 100614.94 | 101219.68 |
| TR | 184267.47 | 185347.15 | 184477.5 |
| TCO | 39.0 | 39.0 | 39.0 |
| TCSS | 18.0 | 18.0 | 18.0 |
| TCP | 78156.75 | 75156.64 | 77727.7 |
| TCI | 246.46 | 315.61 | 273.5 |
| TCS | 3216.75 | 3452.03 | 3513.0 |
| TCU | 426.64 | 416.15 | 466.8 |
| TCLS | 1625.01 | 5334.78 | 1219.5 |

The best values of the objective function and computational time for five different runs of the small-sized test problem are given in Table 8.

In the following, the other variables and related costs from the solving of the small-sized test problems by the GA and VDO algorithms. Table 9 and Fig 10 display the costs of designing this system with different solving methods.

We observe that in solving the small-sized problems the gap for the GA is 0.6726% and for the VDO algorithm is 0.5974%. The computational time is 4.03 seconds for the GA and 2.37 for the VDO algorithm. Thus, these algorithms can be employed in solving larger and real problems.

**5.3.2 Large-sized test problem.** In this subsection, we evaluate the efficiency of the two algorithms in solving large-sized test problems. For this, we designed and solved 15 problems of different sizes (Table 10) by the GA and VDO algorithm.

The averages of the sales profit and computational time are reported in Table 11. Besides, Figs 11 and 12 display the behavior of the objective functions and computational time for these problems solved by these two algorithms.

Fig 12 shows that the average computational time has increased exponentially by the size of the problem. This demonstrates the problem complexity (NP-hardness). Besides, the average objective functions for the different test problems are close to each other (with a slight difference) for both algorithms. However, we should select the most efficient algorithm when solving a real problem. For this, we first test the significance of the difference between the average objective functions and computational times in the 15 test problems at confidence level 0.95. A statistic score lower than 0.05 identifies the significance of the differences. Table 12 reports the results of the t-test performed for this purpose.

According to Table 12, the average profit for the large-sized problems for the GA is greater than that for the VDO algorithm, while the average computational time is smaller for this algorithm. Besides, the results of the t-test indicate that there is no significant difference between

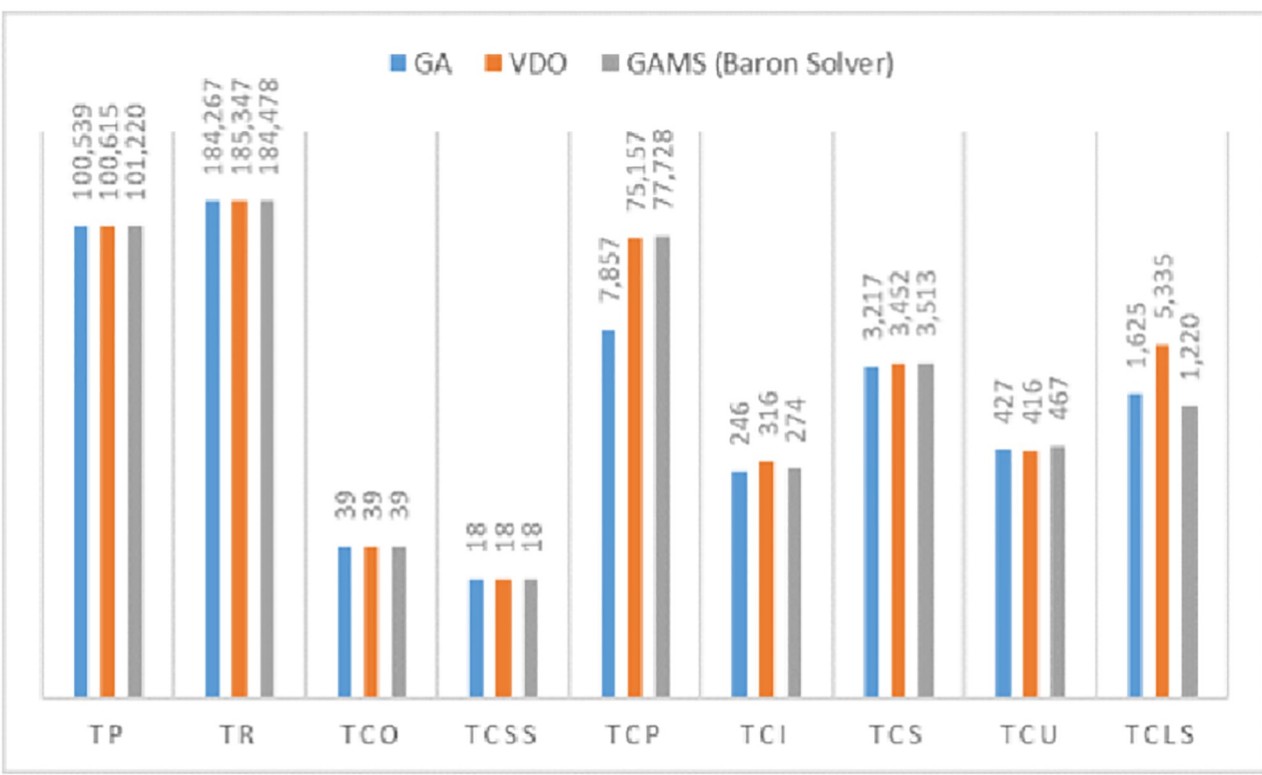

**Fig 10. The comparison of the costs and incomes of designing the system with different solving methods.**

the average objective functions and computational times of these two algorithms. Thus, to select the more efficient algorithm to be used in solving large-sized problems, we used the TOPSIS method, which a multi-criteria decision making method. By considering the weight of 0.8 for the objective function and 0.2 for the computational time, the results indicate the selection of the VDO algorithm with utility weight 0.6986.

**Table 10. The size of the designed test problems.**

| Sample Problem | I = K | T | G | J |
|:---:|:---:|:---:|:---:|:---:|
| 1 | 6 | 6 | 6 | 4 |
| 2 | 8 | 6 | 6 | 4 |
| 3 | 10 | 6 | 7 | 5 |
| 4 | 12 | 8 | 7 | 5 |
| 5 | 14 | 8 | 7 | 5 |
| 6 | 16 | 8 | 8 | 6 |
| 7 | 18 | 8 | 8 | 6 |
| 8 | 20 | 10 | 8 | 6 |
| 9 | 23 | 10 | 8 | 7 |
| 10 | 27 | 10 | 9 | 7 |
| 11 | 30 | 10 | 9 | 8 |
| 12 | 35 | 12 | 10 | 8 |
| 13 | 40 | 12 | 10 | 9 |
| 14 | 45 | 14 | 12 | 9 |
| 15 | 50 | 15 | 12 | 10 |

**Table 11. The averages of the objective function and computational time in solving the large-sized problems by metaheuristic algorithms.**

| Problem Instances | VDO | | GA | |
|---|---|---|---|---|
| | Objective function | Cpu-Time (sec) | Objective function | Cpu-Time (sec) |
| 1 | 174243.31 | 9.97 | 179916.48 | 11.95 |
| 2 | 242026.93 | 13.28 | 232210.80 | 18.93 |
| 3 | 270094.30 | 19.60 | 275238.83 | 22.63 |
| 4 | 328096.73 | 27.39 | 333150.11 | 32.29 |
| 5 | 376288.80 | 38.66 | 393615.39 | 45.30 |
| 6 | 437334.81 | 53.08 | 457229.13 | 61.17 |
| 7 | 470518.60 | 68.71 | 480430.59 | 81.13 |
| 8 | 515879.02 | 85.95 | 554769.85 | 103.20 |
| 9 | 560557.88 | 105.32 | 568024.45 | 129.89 |
| 10 | 613858.83 | 124.47 | 625099.28 | 157.77 |
| 11 | 685489.81 | 153.90 | 716325.35 | 191.91 |
| 12 | 715829.38 | 182.44 | 746326.12 | 228.60 |
| 13 | 795440.00 | 210.83 | 767983.45 | 268.66 |
| 14 | 830562.89 | 243.28 | 824861.61 | 311.22 |
| 15 | 871235.85 | 278.33 | 924757.91 | 359.19 |

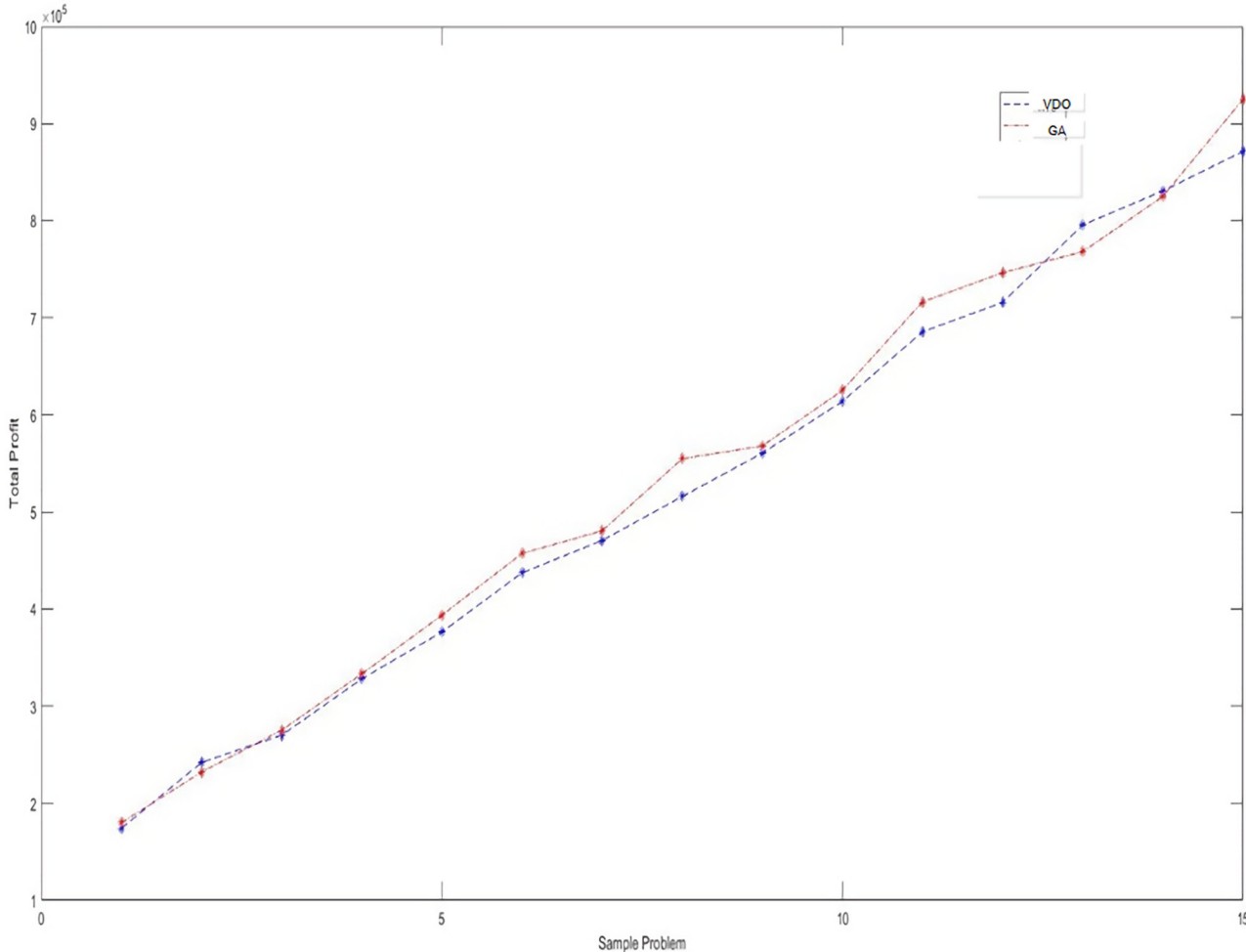

**Fig 11. The trend of the objective function for the large-sized test problems.**

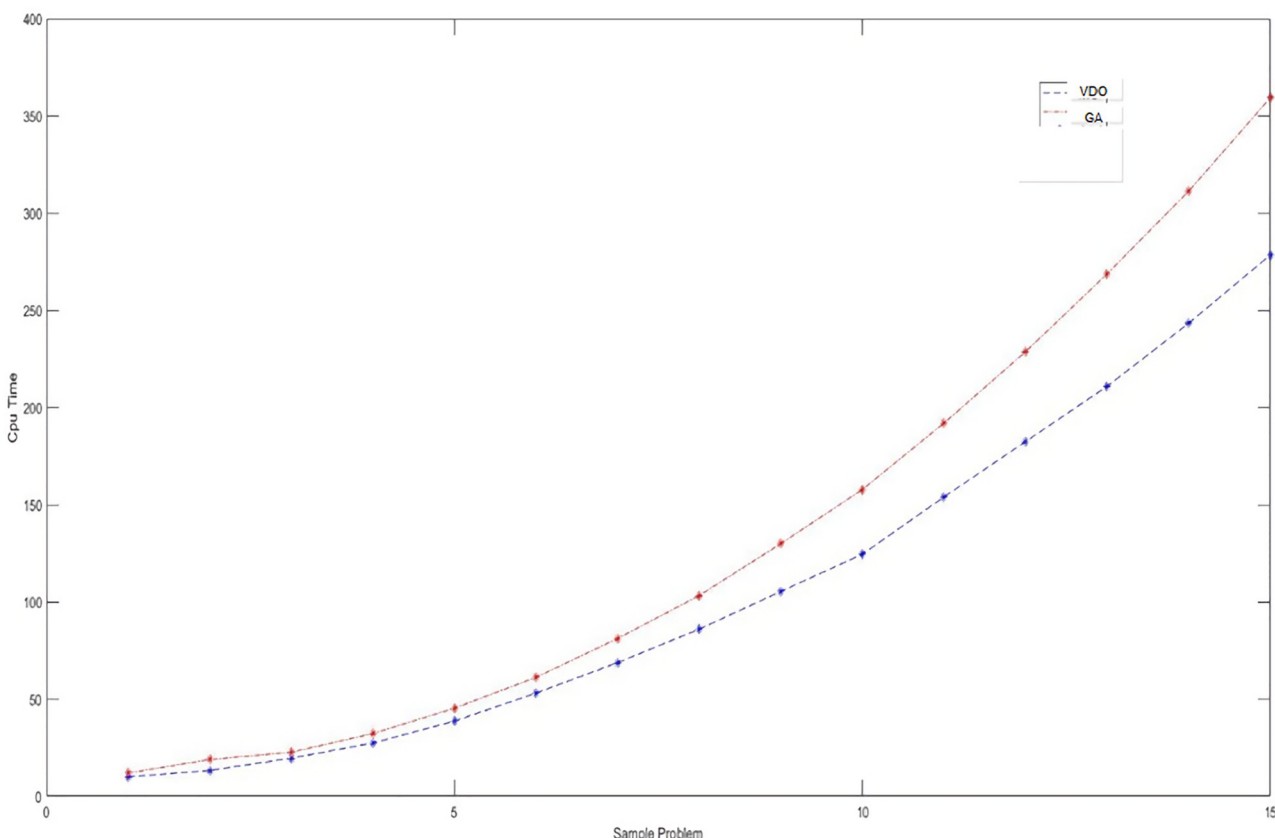

**Fig 12. The trend of computational time for the large-sized test problems.**

## 5.4. A real case study

To implement and analyze the developed model, we use the data of a real case in Iran. Ofogh Koorosh chain stores with 14000 personnel and 2180 store established to provide fast delivering products through grocery retailing. They offer agility and speed to serve as the basis for creating a dynamic and stable competitive advantage and based their competitive strategy on strict adherence to business ethics. They provide permanent discounts on all products as compared to the price labeled on the product, where these products are mainly used for daily, weekly, or monthly consumption of the Iranian household. We considered for our problem the data for dairy products consisting of 423 items, supplied by 37 suppliers, in a 30-day interval.

**Table 12. The results of T-test for the significance testing of the differences between the considered criteria in solving large-sized problems.**

| Algorithms | criteria | Mean | Estimate for difference | 95% CI for the difference | T-Value | P-Value |
|---|---|---|---|---|---|---|
| VDO | Objective Function | 525830 | 12832 | (-156459, 182123) | 0.16 | 0.878 |
| GA | | 538663 | | | | |
| VDO | Cpu-time | 107.7 | 27.2 | (-49.1, 103.6) | 0.73 | 0.470 |
| GA | | 135 | | | | |

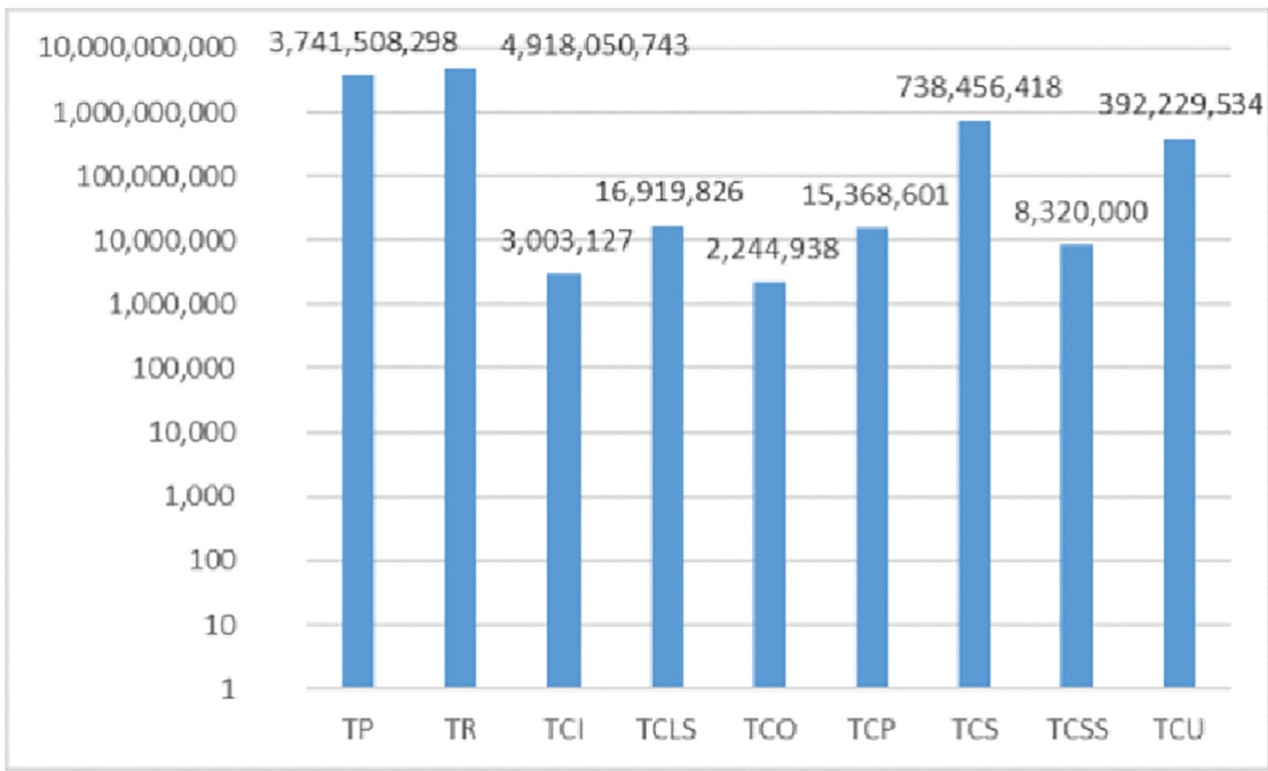

**Fig 13. The costs and total profit of the real case problem.**

The VDO algorithm solved the problem in 5795.24 seconds and reached a profit as large as 3741508298.35 in 150 consecutive iterations. In the obtained solution, only suppliers 9, 12, 15, 35, and 37 were not selected. Fig 13 displays the other related costs and Fig 14 displays the convergence of the VDO algorithm in getting the maximum profit for this real problem.

Besides, Figs 15 and 16 shows the shelf average inventory for each group of products for 30 days. According to this figure, the most inventory is for milk items.

Table 13 gives the results for each product group in the real problem, where the most purchased items belong to cheese and after that to high-fat yogurt and low-fat milk. The most display at each period has been allocated on average to cheese group of products and after that to high-fat yogurt and flavored milk. The most shelf-space at each period has been allocated on average to cheese, high-fat yogurt, and flavored milk groups.

## 6. Conclusions

In this research, we considered a retailing problem seeking for maximizing its profit from the sales of the principal or substituting products. This problem considers the perishability of the products and it incurs a cost for destroying the products if they are not sold. Before we solved large-sized problems, we analyzed the problem in small-sized cases, whose results indicated the high computational time of the problem by the exact methods. The sensitivity analysis showed that changes in the product demand, transportation cost, and price had a direct impact on the total profit. Due to the failure of exact methods in solving large-sized problems, we used

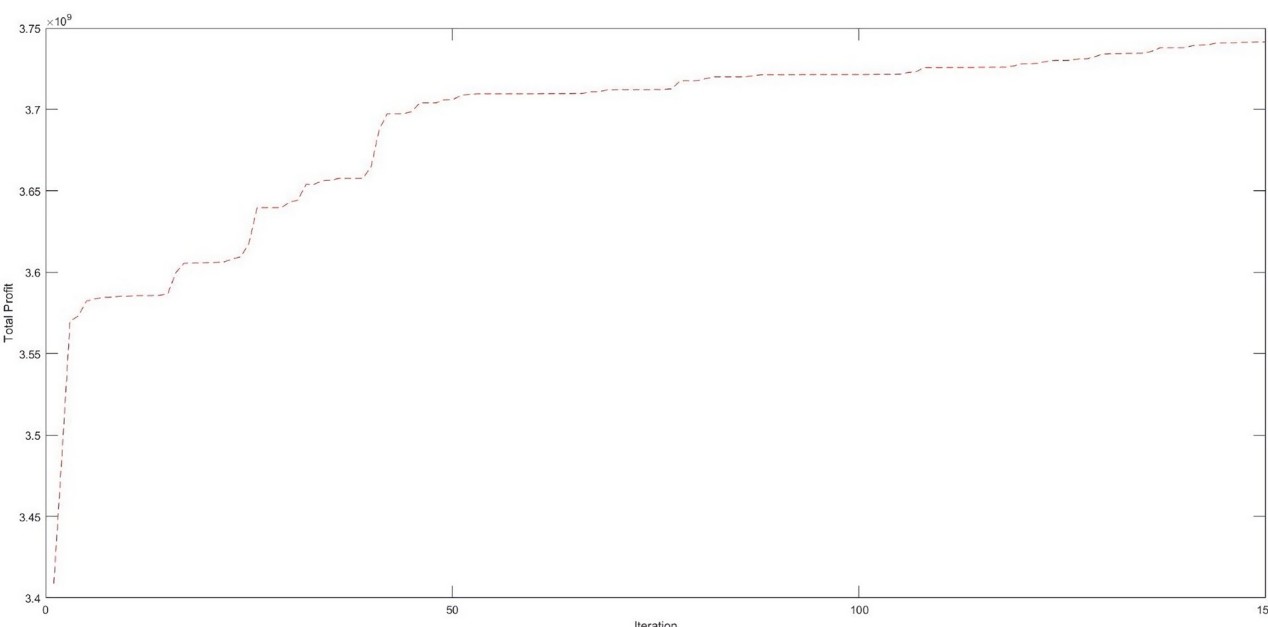

**Fig 14. The convergence of the VDO algorithm in reaching the maximum profit.**

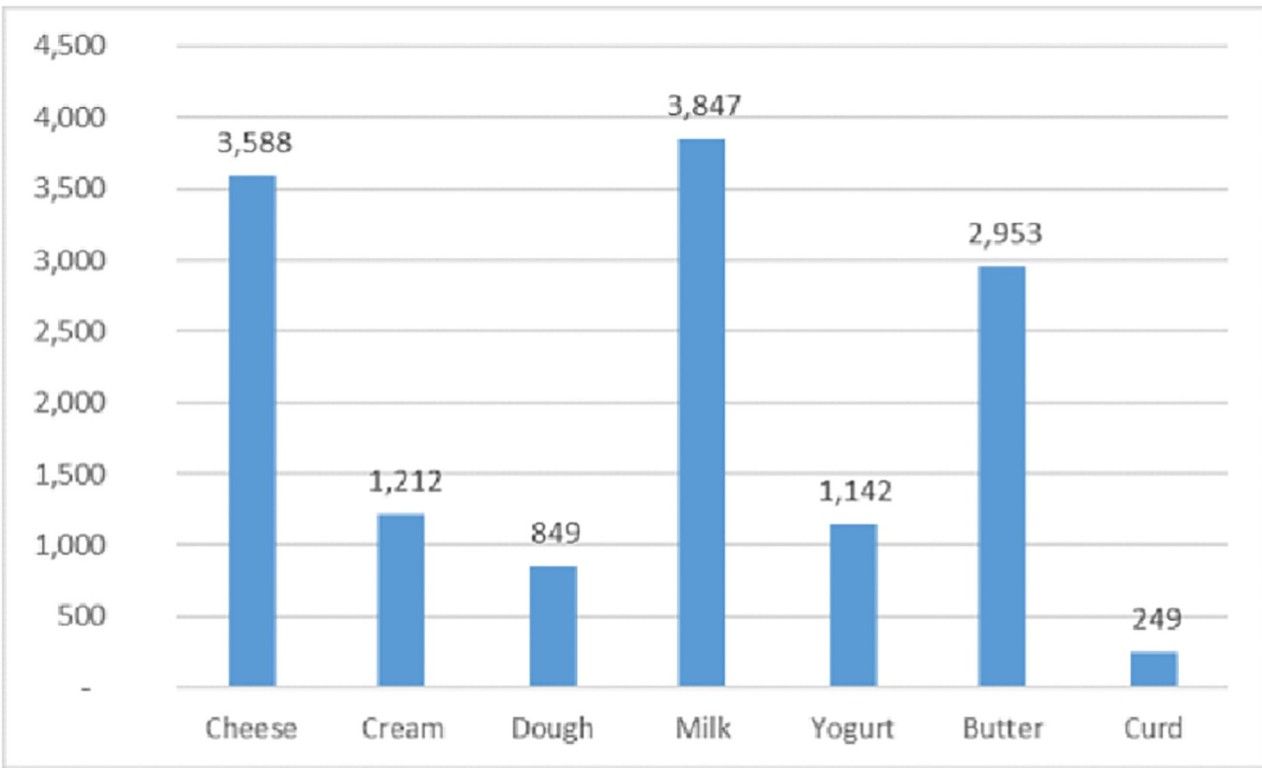

**Fig 15. The average inventory for the products in the real problem.**

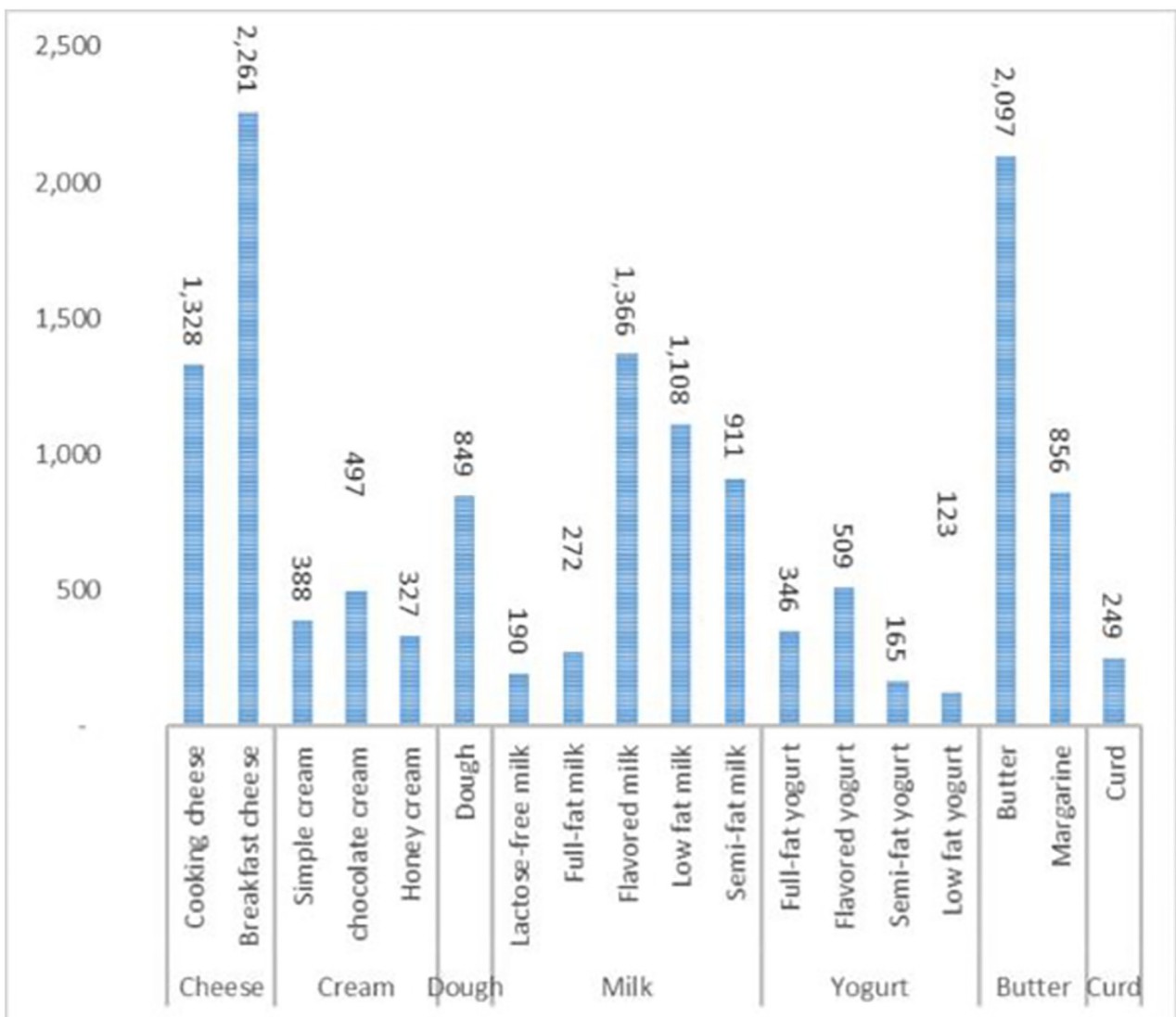

**Fig 16. The average inventory for the products in the real problem.**

two metaheuristic algorithms, GA and VDO. The results of these algorithms for small-sized problems verified the efficiency of both algorithms in obtaining near-optimal solutions in a short time, such that the maximum relative difference was less than one percent. To select the most efficient algorithm, we designed 15 test problems of different sizes and solved by the two algorithms. The results of the statistical test on the averages of the objective function and computational time showed that there was no significant difference between these two criteria. Therefore, we used the TOPSIS method for the selection of the most efficient algorithm, indicating the superiority of the VDO algorithm in solving large-sized problems. To implement this problem in a real world case, we considered Ofogh Koorosh chain stores in Iran. The results confirmed the applicability of the proposed model and metaheuristic algorithms.

The model proposed by this study has several opportunities for expansion, first it is assumed that the usual demand of each product is known in each period. However, this detail

**Table 13. The results of the real problem solved by VDO algorithm.**

| Product groups | Product availability | Number of selected | Number of supplier | Average of periodic demand | Average of display number | Average of shelf space allocation |
|---|---|---|---|---|---|---|
| Cooking cheese | 20 | 8 | 18 | 63 | 13 | 1,889 |
| Breakfast cheese | 86 | 38 | 84 | 210 | 60 | 7,906 |
| Simple cream | 13 | 7 | 13 | 127 | 12 | 1,113 |
| Chocolate cream | 4 | 3 | 4 | 47 | 4 | 282 |
| Honey cream | 6 | 4 | 6 | 26 | 7 | 497 |
| Dough | 44 | 20 | 44 | 311 | 34 | 3587 |
| Lactose-free milk | 1 | 1 | 1 | 4 | 1 | 76 |
| Full-fat milk | 35 | 21 | 35 | 202 | 29 | 2,889 |
| Flavored milk | 39 | 13 | 36 | 169 | 34 | 3,317 |
| Low fat milk | 38 | 24 | 38 | 335 | 30 | 3,098 |
| Semi-fat milk | 2 | 0 | 2 | 61 | 2 | 192 |
| Butter | 24 | 11 | 23 | 86 | 17 | 2,173 |
| Margarine | 3 | 1 | 3 | 3 | 2 | 288 |
| Curd | 4 | 2 | 4 | 1 | 5 | 333 |
| Full-fat yogurt | 53 | 25 | 53 | 248 | 37 | 4,630 |
| Flavored yogurt | 12 | 7 | 12 | 42 | 9 | 1,053 |
| Low fat yogurt | 36 | 17 | 36 | 204 | 27 | 3,256 |
| Semi-fat yogurt | 3 | 3 | 3 | 12 | 3 | 272 |

is unknown and uncertain. Therefore, considering the uncertainty in demand could significantly improves the retailer' planning. Second, another important decision for category managers in retail industries is pricing. Seting price of each product included in the assortment effect on other retail category management decision aspects such as assortment planning, shelf space allocation and inventory planning. Therefore introducing a model that addressed all aspect a retail.

Third, COVID-19 pandemic has had a huge impact on the retail industry, Social distancing has become commonplace, and the world's leading retailers are using some of their offline stores as dark stores, so implementing a model for assortment planning, shelf space allocation and inventory management in Omni channel retailing can be considered. Also, developing a multi-objective model with sale losses and lead time consideration can be attractive topics for future research.

## Supporting information

**S1 Data.**
(XLSX)

## Author Contributions

**Investigation:** Ali Ahmadi.

**Supervision:** Seyed Jafar Sajadi.

**Writing – original draft:** Ali Ahmadi.

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
