## [Decision Letter · Decision Letter 0]

4 Nov 2021

PONE-D-21-28756An integrated optimization model and metaheuristics for assortment planning, shelf space allocation, and inventory management of perishable products: A real applicationPLOS ONE

Dear Dr. Ahmadi,

Thank you for submitting your manuscript to PLOS ONE. After careful consideration, we feel that it has merit but does not fully meet PLOS ONE’s publication criteria as it currently stands. Therefore, we invite you to submit a revised version of the manuscript that addresses the points raised during the review process.

We look forward to receiving your revised manuscript.

Kind regards,

Yong Wang

Academic Editor

PLOS ONE

Journal Requirements:

Additional Editor Comments (if provided):

Dear Authors,

Please see the reviewers' comments and revise your paper, thanks.

Best Wishes,

Yong Wang, Ph.D.

Professor,

School of Economics and Management,

Chongqing Jiaotong University, Chongqing, China

Reviewers' comments:

Reviewer's Responses to Questions

**Comments to the Author**

1. Is the manuscript technically sound, and do the data support the conclusions?

Reviewer #1: Partly

Reviewer #2: Yes

2. Has the statistical analysis been performed appropriately and rigorously? 

Reviewer #1: No

Reviewer #2: Yes

3. Have the authors made all data underlying the findings in their manuscript fully available?

Reviewer #1: Yes

Reviewer #2: Yes

4. Is the manuscript presented in an intelligible fashion and written in standard English?

Reviewer #1: No

Reviewer #2: Yes

5. Review Comments to the Author

Reviewer #1: Attached please download the review report. Thank you.

Review report

This study investigated an integrated optimization problem considering the assortment planning, shelf space allocation, and inventory management and supplier selection for the perishable products. A mixed integer non-linear programming model was established to formulate the NP-hard problem. An evolutionary genetic algorithm and vibration damping optimization algorithm were employed to solve the proposed model. Numerical experiments and a real-life case study were exemplified to validate the proposed model and solution methods. Overall, the authors have spent a lot of efforts on this study. However, the current version of this paper has a number of limitations that preclude its publication. In what follows we are identifying the major shortcomings of the paper:

1. There are several errors or unclear expressions in the model formulation.

1) The authors should use a unified expression (e.g., expressed by using Mathtype software or formula editor embedded in Microsoft Office) to type the equations. The inconsistence appeared in both the model formulation section and the case study section. The authors should revise the corresponding parts.

2) Some typos are not defined, such as I, J, G, x_it, s_k, V_it^G… Do I/J/G represent a number or set?

3) The author defined many types of cost; however, the difference of these cost was not clearly illustrated. For example, what is the selecting cost and what’s the difference between it and the fixed cost of ordering from suppliers? What’s the difference of the penalty cost of substitution for product i and the unit cost of non-responding to the demand of product i? The authors should give more detailed explanation for these costs to show readers with concrete understanding.

4) Some definitions are difficult to understand. For example, m_ikt^g is defined as the amount of product i with lifetime g allocated to the demand of product k at time t. Does it represent the amount of product i with lifetime g that substituted by the product k at time t? If yes, only this sales income was calculated? How about the sales income of directly selling the product i? The β_i expressed the spatial elasticity of product i. But there is only β appeared in equation (17). A lot of these inconsistences appears in the modeling part. The authors should proofread the whole mathematical formulation to ensure right and consistent expressions, meanwhile, give more detailed explanations or a specific example to illustrate some abstract equations, such as β_i.

5) Why is there a 2 in the denominator for the calculation of TCI in equation (5)?

6) It is better to point out the specific set that the variable belongs to when conducting the summation to show a clearer expression. For example, TCO=∑_(j∈J)▒∑_tϵT▒〖〖oc〗_j∙o_jt 〗 instead of TCO=∑_j▒∑_t▒〖〖oc〗_j∙o_jt 〗.

2. There is no need to show all details for the encoding and decoding procedure. Only an example for the same or similar process is enough because several parts are similar. Additionally, the figure is not clear with blurry expression.

3. All the parameter settings and the information of the computer or workstation used for calculation should be fully illustrated in an appropriate place.

4. Some results and analysis may be wrong. For example, in Table 9, the value of TP illustrated in the second row that calculated by GA and GAMS is not consistent with the value that directly calculated by equation (1). Similar problems are appeared in Tables 6 and 7. The illustration of the gaps for GA and VDO under figure 10 is 0.6726% and 0.5974% instead of 0.6726 and 0.5974. The authors should take care of these problems and ensure accurate and authentic results.

5. What is the significance and specification of considering the perishable products instead of other products on model formulation or to the practical implication. Is the consideration of lifetime g for perishable products? If yes, a sensitive analysis for the lifetime should also be conducted.

Reviewer #2: PLOS ONE

Manuscript Draft

Manuscript Number: PONE-D-21-28756

Title: An integrated optimization model and metaheuristics for assortment planning, shelf space allocation, and inventory management of perishable products: A real application

*Comments to the Author:

In this paper, an integrated optimization model and metaheuristics for assortment planning, shelf space allocation, and inventory management of perishable products are investigated using an evolutionary genetic algorithm (GA), and an efficient local search vibration damping optimization (VDO) algorithm. The results basically consist of the comparison of the proposed approach with the GA and VDO algorithm. An empirical case study is conducted and the effectiveness of the proposed formulation and the applicability of two algorithms for various instances are evaluated.

Overall, the manuscript has been prepared in a professional manner. It is well organized and the message is clear. However, there are areas for improvement in terms of content. The content requires more scientific support. The current manuscript has a major revision of some issues that need to be addressed before being considered for publication.

1. What are some ways in which the paper could be improved? Please supply any additional important references that you feel the author omitted which should be noted in the paper. First, the problem presented in this work is formulated as a single objective function. I would suggest authors considering a separate objective for the maximize sales problem.

2. The author’s research is only the integration of a single research element in the existing literature, and does not highlight the intention of the problem. It is suggested that the author supplement it.

3. The literature review part lacks a detailed review of the model and methodology. It is suggested to refer to the following papers:

[1] Wang, Y., Peng, S. G., Zhou, X. S., Mahmoudi, M., Zhen, L. Green logistics location-routing problem with eco-packages. Transportation Research Part E: Logistics and Transportation Review, 2020, 143, 102118.

[2] Wang, Y., Yuan, Y. Y., Guan, X., Y., Xu, M. Z., Wang, L., Wang, H. Z., Liu, Y. Collaborative two-echelon multicenter vehicle routing optimization based on state–space–time network representation. Journal of Cleaner Production, 2020, 258, 120590.

[3] Wang, Y., Peng, S. G., Xu, M. Emergency logistics network design based on space–time resource configuration. Knowledge-Based Systems, 2021, 223, 107041.

[4] Wang, Y., Zhang, S. L., Guan, X. Y., Fan, J. X., Wang, H. Z., Liu, Y. Cooperation and profit allocation for two-echelon logistics pickup and delivery problems with state–space–time networks. Applied Soft Computing, 2021, 109, 107528.

4. The authors mentioned “To overcome these drawbacks, this paper develops an integrated mix-integer nonlinear mathematical model”. How does the author deal with the nonlinear programming model in this paper?

5. Why does the authors not consider the combination of any two of the three factors in the paper?

6. In the sensitivity analysis, why does the authors not consider the impact of inventory on retailers' revenue and sales?

7. The authors need to provide a practical example to prove that the research problem originates from practice and assist readers in understanding this research.

8. Some more valuable managerial insights should be provided by sensitive analysis.

9. In terms of result comparison, the authors only compare the algorithm proposed in this paper with the results obtained by gams solver. It is suggested that the authors supplement the algorithm comparison results in other literature.

10. Punctuation is missing in some parts of the article. The author suggested checking the whole paper.

For example, “ To overcome these drawbacks this paper develops an integrated mix-integer nonlinear mathematical model for assortment planning, supplier selection, shelf space allocation, and inventory management.”

6. PLOS authors have the option to publish the peer review history of their article (what does this mean?). If published, this will include your full peer review and any attached files.

Reviewer #1: No

Reviewer #2: No

---

## [Author Response · Author response to Decision Letter 0]

4 Dec 2021

Response to Reviewers' Comments on PONE-D-21-28756] - [EMID: 8573feb3eaa1f7ee]

An integrated optimization model and metaheuristics for assortment planning, shelf space allocation, and inventory management of perishable products: A real application

 The authors would like to acknowledge and thank the referees and the editor for their profoundly insightful comments, which enabled us to significantly improve the quality of our manuscript. The following pages contains our point-by-point responses to the comments. The revision has been made in consultation with all authors, and each author has approved the final form of this revision. Another version of the document without any markings has been included, as well.

 

Reviewer #1: PLOS ONE

Manuscript Draft

Manuscript Number: PONE-D-21-28756

This study investigated an integrated optimization problem considering the assortment planning, shelf space allocation, and inventory management and supplier selection for the perishable products. A mixed integer non-linear programming model was established to formulate the NP-hard problem. An evolutionary genetic algorithm and vibration damping optimization algorithm were employed to solve the proposed model. Numerical experiments and a real-life case study were exemplified to validate the proposed model and solution methods. Overall, the authors have spent a lot of efforts on this study. However, the current version of this paper has a number of limitations that preclude its publication. In what follows we are identifying the major shortcomings of the paper:

We appreciate that the reviewer’s comments. The followings are our point-by-point responses:

 There are several errors or unclear expressions in the model formulation.

1) The authors should use a unified expression (e.g., expressed by using Mathtype software or formula editor embedded in Microsoft Office) to type the equations. The inconsistence appeared in both the model formulation section and the case study section. The authors should revise the corresponding parts.

Of course, we unified all formulations in Microsoft Word form and excluded the MathType configuration. A sample of it is as follows:

(22) ∀i,j,t x_it.a_ij≤M.o_jt

(23) ∀i,t f_it≤M〖yy〗_it

(24) ∀i,t f_it≥〖yy〗_it

2) Some typos are not defined, such as I, J, G, x_it, S_k, V_it^G… Do I/J/G represent a number or set?

We explained all indexes in the 3.2 section for example:

i Index of the product number

j Index of supplier number

g Index of product lifetime

They were represented sets to use in modeling. However, we changed the first section of the indices table header to sets, which will help to increase the readers' understanding. 

In the following, we presented x_i^t and V_it^g definitions as variables:

x_i^t Required amount of product i at time t

V_it^g Amount of inventory of product i with lifetime g at the end of period t

We had a typo in the S definition; the correct definition is as follows:

s_k Penalty cost of substitution for product k

3) The author defined many types of cost; however, the difference of these cost was not clearly illustrated. For example, what is the selecting cost and what’s the difference between it and the fixed cost of ordering from suppliers? What’s the difference of the penalty cost of substitution for product i and the unit cost of non-responding to the demand of product i? The authors should give more detailed explanation for these costs to show readers with concrete understanding.

This comment is precise; this part may be confusing for readers. In this way, we added a new paragraph to explain these costs. It is as follows:

“Where supplier selection costs explain contract registration costs and ordering costs show supplying products costs from a specific supplier. Another hand, the penalty cost for substituting is a cost based on creating customer distrust. The penalty cost of not responding to the demands represents demand loss cost.”

4) Some definitions are difficult to understand. For example, m_ikt^G% is defined as the amount of product i with lifetime g allocated to the demand of product k at time t. Does it represent the amount of product i with lifetime g that substituted by the product k at time t? If yes, only this sales income was calculated? How about the sales income of directly selling the product i? The β_i expressed the spatial elasticity of product i. But there is only β appeared in equation (17). A lot of these inconsistences appears in the modeling part. The authors should proofread the whole mathematical formulation to ensure right and consistent expressions, meanwhile, give more detailed explanations or a specific example to illustrate some abstract equations, such as β_i.

Yes, m_ikt^G represents the amount of product i with lifetime g that is substituted by the product k at time t. Of course, we have a typo in this equation that after correction appeared as follows:

TR=∑_i▒∑_k▒∑_g▒∑_t▒〖p_i.〖(m〗_ikt^g+x_i^t)〗

In this form, we calculated both sales income for products substituted and direct sales.

Existed β_i to equation (17) is related to the Total demand variable (d_i^t) and the number of displays that can be allocated to products variable (f_it). Another hand, these variables are linked to other variables by equations (14,15,16,18,23,24).

The proposed model is an integrated model which can create a strong relationship between all variables. Finally, we investigated all equations in the mathematical model.

5) Why is there a 2 in the denominator for the calculation of TCI in equation (5)?

The inventory variables definition implied the inventory level at the end of period t; for this reason, we calculated an average inventory level.

We added “average” at the beginning of the description as follows:

“Eq.s (2)-(9) show respectively the amount of sales income, fixed cost of ordering, cost of supplier selection, purchase costs, average inventory holding costs, penalty cost for substituting another product instead of the desired product, cost incurred due to the expired date and the corruption of the product, and penalty cost of not responding to the demands.”

6) It is better to point out the specific set that the variable belongs to when conducting the summation to show a clearer expression. For example, TCO=∑_jϵJ▒∑_tϵT▒〖〖oc〗_j.o_jt 〗instead of TCO=∑_j▒∑_t▒〖〖oc〗_j.o_jt 〗.

We corrected all inconsistency of this kind in all over text. For example,

(6) TCI=∑_(i∈I)▒∑_(g∈G_i)▒∑_(t∈T)▒〖(V_it^g)/2.h_i 〗

2. There is no need to show all details for the encoding and decoding procedure. Only an example for the same or similar process is enough because several parts are similar. Additionally, the figure is not clear with blurry expression.

We summarized the procedures of our metaheuristic approach in Fig. 1. We also increased the sharpness and quality of the figure by preserve details 2.0 (AI method) as follows:

3. All the parameter settings and the information of the computer or workstation used for calculation should be fully illustrated in an appropriate place.

We added a new paragraph in section 5 that explains the workstation system features and presented parameter values setting in an appropriate section. It is as follows:

“The proposed mathematical model has a profit objective function for assortment planning, shelf space allocation, and inventory management of perishable products. It is solved with the GAMS 24.1.2 (BARON Solver) and Matlab 2019(b) software using a core i5 CPU (2GHz frequency), 8 GB RAM processor. Table 1 displays the parameters of a small-sized problem.”

4. Some results and analysis may be wrong. For example, in Table 9, the value of TP illustrated in the second row that calculated by GA and GAMS is not consistent with the value that directly calculated by equation (1). Similar problems are appeared in Tables 6 and 7. The illustration of the gaps for GA and VDO under figure 10 is 0.6726% and 0.5974% instead of 0.6726 and 0.5974. The authors should take care of these problems and ensure accurate and authentic results.

It isn't inconsistent because we conducted our proposed approach on two cases, including a sample case instance and a case of the DRCFJSS problem.

In section 5.3, we mentioned "evaluating the proposed GA and VDO solution approaches, the case of the DRCFJSS problem is solved using these algorithms, and their results are compared with the global optimum solution obtained by GAMS." Also, sections 5.1 and 5.2 are implemented on a small-scale problem where is mentioned: "In this section, we solve a small-scale problem."

it was corrected as follows:

“We observe that in solving the small-sized problems the gap for the GA is 0.6726% and for the VDO algorithm is 0.5974%.”

5. What is the significance and specification of considering the perishable products instead of other products on model formulation or to the practical implication. Is the consideration of lifetime g for perishable products? If yes, a sensitive analysis for the lifetime should also be conducted.

In the proposed model, we defined a set for lifetime products, including an index of the lifetime for each product. This feature allows us to determine a specific lifetime for each product and not a general version for all products. In other words, the relevant variables take the value zero for out of lifetime cases. 

Determining this feature is a unique index for perishable products that aren't seen in other products.

According to different values lifetime of products, we conducted a sensitivity analysis for lifetime products with changing current values of our products. It was added to the main text as follows:

 “Lifetime SA

Figure 8 shows the behaviors of the changes in the sales profits due to the changes in the current lifetime products. Generally, It is observed that the amount of sales profits has increased by the increase of the lifetime. Sales profit decreases with a steeper slope for reducing product lifetime due to growing perishable costs. Sales profits have increased by the rise in the lifetime, but it has a lower pitch because the inventory costs are growing.”

Figure 8- The trend of changes in the sales profits due to the changes in the lifetime

  

Reviewer #2: PLOS ONE

Manuscript Draft

Manuscript Number: PONE-D-21-28756

Title: An integrated optimization model and metaheuristics for assortment planning, shelf space allocation, and inventory management of perishable products: A real application

*Comments to the Author:

In this paper, an integrated optimization model and metaheuristics for assortment planning, shelf space allocation, and inventory management of perishable products are investigated using an evolutionary genetic algorithm (GA), and an efficient local search vibration damping optimization (VDO) algorithm. The results basically consist of the comparison of the proposed approach with the GA and VDO algorithm. An empirical case study is conducted and the effectiveness of the proposed formulation and the applicability of two algorithms for various instances are evaluated.

Overall, the manuscript has been prepared in a professional manner. It is well organized and the message is clear. However, there are areas for improvement in terms of content. The content requires more scientific support. The current manuscript has a major revision of some issues that need to be addressed before being considered for publication.

We appreciate that the reviewer’s comments. The followings are our point-by-point responses:

1. What are some ways in which the paper could be improved? Please supply any additional important references that you feel the author omitted which should be noted in the paper. First, the problem presented in this work is formulated as a single objective function. I would suggest authors considering a separate objective for the maximize sales problem.

Sale losses minimization can be an attractive topic as an influential factor on the incomes system. Also, lead time minimization is another critical issue that can be discussed in future studies. Where we mentioned in conclusion section as follows:

Third, COVID-19 pandemic has had a huge impact on the retail industry, Social distancing has become commonplace, and the world's leading retailers are using some of their offline stores as dark stores, so implementing a model for assortment planning, shelf space allocation and inventory management in Omni channel retailing can be considered. Also, developing a multi-objective model with sale losses and lead time consideration can be attractive topics for future research.

To complete the references, we added some studies as follow:

32. Hübner, A., Kuhn, H. and Kühn, S., 2016. An efficient algorithm for capacitated assortment planning with stochastic demand and substitution. European Journal of Operational Research, 250(2), pp.505-520

 Hübner, A., 2017. A decision support system for retail assortment planning. International Journal of Retail & Distribution Management.

In connection with the last part of your comment, we proposed a profit objective function that shows the subtraction of revenues from expenses. This function tries to maximize sales automatically because revenue from sales has appeared with a positive sign.

2. The author’s research is only the integration of a single research element in the existing literature, and does not highlight the intention of the problem. It is suggested that the author supplement it.

To better explain the issue, We added three paragraphs in the problem statement section that has explained assortment, shelf-space, and inventory management problems. They were presented as follow:

“Assortment planning is implied to the set of decisions for products carried in each store at each point in time. The target of assortment planning optimization is to determine an assortment that maximizes sales or gross subject to various constraints, such as a limited budget for purchase of products, limited shelf space for displaying products, and a variety of multiple constraints such as a desire to have at least two vendors for each type of product.

Shelf-space mathematical models optimize the number of facings for items with space-elastic demand to be allocated to limited shelf space. Respective approaches aid retailers in dealing with the trade-off between more shelf space (and thus demand enhancement due to a higher number of facings) for specific items and less available space (and therefore demand decreases due to a lower number of facings) for other products.

Multi-item inventory problems are also highly relevant to the assortment planning problem. The inventory management of multiple products under shelf space limitations or budget constraints can be a critical issue that needs consideration.”

3. The literature review part lacks a detailed review of the model and methodology. It is suggested to refer to the following papers:

[1] Wang, Y., Peng, S. G., Zhou, X. S., Mahmoudi, M., Zhen, L. Green logistics location-routing problem with eco-packages. Transportation Research Part E: Logistics and Transportation Review, 2020, 143, 102118.

[2] Wang, Y., Yuan, Y. Y., Guan, X., Y., Xu, M. Z., Wang, L., Wang, H. Z., Liu, Y. Collaborative two-echelon multicenter vehicle routing optimization based on state–space–time network representation. Journal of Cleaner Production, 2020, 258, 120590.

[3] Wang, Y., Peng, S. G., Xu, M. Emergency logistics network design based on space–time resource configuration. Knowledge-Based Systems, 2021, 223, 107041.

[4] Wang, Y., Zhang, S. L., Guan, X. Y., Fan, J. X., Wang, H. Z., Liu, Y. Cooperation and profit allocation for two-echelon logistics pickup and delivery problems with state–space–time networks. Applied Soft Computing, 2021, 109, 107528.

These are precious studies that we added to our references as follow:

34. Wang, Y., Peng, S. G., Zhou, X. S., Mahmoudi, M., Zhen, L. Green logistics location-routing problem with eco-packages. Transportation Research Part E: Logistics and Transportation Review, 2020, 143, 102118.

35. Wang, Y., Yuan, Y. Y., Guan, X., Y., Xu, M. Z., Wang, L., Wang, H. Z., Liu, Y. Collaborative two-echelon multicenter vehicle routing optimization based on state–space–time network representation. Journal of Cleaner Production, 2020, 258, 120590.

36. Wang, Y., Peng, S. G., Xu, M. Emergency logistics network design based on space–time resource configuration. Knowledge-Based Systems, 2021, 223, 107041.

37. Wang, Y., Zhang, S. L., Guan, X. Y., Fan, J. X., Wang, H. Z., Liu, Y. Cooperation and profit allocation for two-echelon logistics pickup and delivery problems with state–space–time networks. Applied Soft Computing, 2021, 109, 107528

4. The authors mentioned “To overcome these drawbacks, this paper develops an integrated mix-integer nonlinear mathematical model”. How does the author deal with the nonlinear programming model in this paper?

In this way, we explained how to deal with the nonlinear programming model. It related explanation changed as follows:

“The nonlinearity of the demand function makes the problem a mixed-integer non-linear model. GAMS BARON solver is hired to solve the proposed model in small and medium scales. An evolutionary genetic algorithm (GA) and an efficient local search vibration-damping optimization (VDO) algorithm are proposed for large-scale problems.”

5. Why does the authors not consider the combination of any two of the three factors in the paper?

There are several studies implied on the combination of any two of the three factors in the paper that is most important them as follows:

 Pizzi G, Scarpi D. The effect of shelf layout on satisfaction and perceived assortment size: An empirical assessment. J Retail Consum Serv. 2016;28: 67–77.

 Kim G, Moon I. Integrated planning for product selection, shelf-space allocation, and replenishment decision with elasticity and positioning effects. J Retail Consum Serv. 2021;58: 102274.

 Hübner, A., Kuhn, H. and Kühn, S., 2016. An efficient algorithm for capacitated assortment planning with stochastic demand and substitution. European Journal of Operational Research, 250(2), pp.505-520.

 Hübner A, Schaal K. An integrated assortment and shelf-space optimization model with demand substitution and space-elasticity effects. Eur J Oper Res. 2017;261: 302–316.

We observed a research gap in the literature for an integrated study. In this way, the present study proposed an integrated optimization model for assortment planning, shelf space allocation, and inventory management for perishable products.

6. In the sensitivity analysis, why does the authors not consider the impact of inventory on retailers' revenue and sales?

Since inventory costs calculation is formulated as follows: 

TCI=∑_i▒∑_g▒∑_t▒〖(V_it^g)/2.h_i 〗

The inventory variables definition implied the inventory level at the end of period t; for this reason, we calculated an average inventory level for inventory costs.

In this way, the impact of inventory shows lower fluctuation on the sensitivity analysis.

7. The authors need to provide a practical example to prove that the research problem originates from practice and assist readers in understanding this research.

We provided a practical example to validate the research problem in section 5.4 and implemented our proposed mathematical model in a real case (Iran).

We conducted our model in Ofogh Koorosh chain stores with 14000 personnel and 2180 stores established to provide fast-delivery products through grocery retailing.

It could be the best practical example to prove model applicability and assist readers in understanding this research.

8. Some more valuable managerial insights should be provided by sensitive analysis.

We provided more valuable managerial insights by sensitive analysis on products lifetime in the main text as follows:

 “Lifetime SA

Figure 8 shows the behaviors of the changes in the sales profits due to the changes in the current lifetime products. Generally, it is observed that the amount of sales profits has increased by the increase of the lifetime. Sales profit decreases with a steeper slope for reducing product lifetime due to growing perishable costs. Sales profits have increased by the rise in the lifetime, but it has a lower pitch because the inventory costs are growing.”

Figure 8- The trend of changes in the sales profits due to the changes in the lifetime

9. In terms of result comparison, the authors only compare the algorithm proposed in this paper with the results obtained by gams solver. It is suggested that the authors supplement the algorithm comparison results in other literature.

Since there is no integrated optimization model for assortment planning, shelf space allocation, and inventory management for perishable products in the literature, we could not compare our results with other research. Because they generally ignored at least one of these features.

However, to show the proposed approach's better function, we compared our results with the GAMS solver.

10. Punctuation is missing in some parts of the article. The author suggested checking the whole paper.

For example, “ To overcome these drawbacks this paper develops an integrated mix-integer nonlinear mathematical model for assortment planning, supplier selection, shelf space allocation, and inventory management.”

All paragraphs of the main text were checked and corrected based on Punctuation and grammar consideration. All changes can be investigated by the track-change form in Microsoft Word. For example, mentioned paragraph changed to as follows:

“To summarize, assortment planning, space allocation, inventory management, and supplier selection are the most important decisions retailers make, and they are closely related to each other. Demand substitution, space-elasticity demand, and product perishability make these decisions more complicated in the real world. To the best of our knowledge, no studies in the literature have considered all these aspects together. In this way, this paper develops an integrated mix-integer non-linear mathematical model for assortment planning, supplier selection, shelf space allocation, and inventory management. The proposed model considers space elasticity and substitution behavior of customers.

In contrast, in most studies conducted, this study considers perishable products and perishability costs for the retailer. The nonlinearity of the demand function makes the problem a mixed-integer non-linear model. GAMS BARON solver is hired to solve the proposed model in small and medium scales. An evolutionary genetic algorithm (GA) and an efficient local search vibration-damping optimization (VDO) algorithm are proposed for large-scale problems.”

---

## [Decision Letter · Decision Letter 1]

5 Jan 2022

PONE-D-21-28756R1An integrated optimization model and metaheuristics for assortment planning, shelf space allocation, and inventory management of perishable products: A real applicationPLOS ONE

Dear Dr. Ahmadi,

Thank you for submitting your manuscript to PLOS ONE. After careful consideration, we feel that it has merit but does not fully meet PLOS ONE’s publication criteria as it currently stands. Therefore, we invite you to submit a revised version of the manuscript that addresses the points raised during the review process.

We look forward to receiving your revised manuscript.

Kind regards,

Yong Wang

Academic Editor

PLOS ONE

Reviewers' comments:

Reviewer's Responses to Questions

**Comments to the Author**

1. If the authors have adequately addressed your comments raised in a previous round of review and you feel that this manuscript is now acceptable for publication, you may indicate that here to bypass the “Comments to the Author” section, enter your conflict of interest statement in the “Confidential to Editor” section, and submit your "Accept" recommendation.

Reviewer #1: (No Response)

Reviewer #2: All comments have been addressed

2. Is the manuscript technically sound, and do the data support the conclusions?

Reviewer #1: Partly

Reviewer #2: Yes

3. Has the statistical analysis been performed appropriately and rigorously? 

Reviewer #1: No

Reviewer #2: Yes

4. Have the authors made all data underlying the findings in their manuscript fully available?

Reviewer #1: Yes

Reviewer #2: Yes

5. Is the manuscript presented in an intelligible fashion and written in standard English?

Reviewer #1: Yes

Reviewer #2: Yes

6. Review Comments to the Author

Reviewer #1: The authors have already addressed some of my concerns. However, for the question 4 about the inconsistence of the result in Tables 6,7 and 9 still makes me confused. Taking Table 9 as am example, according to the equation (1), TP=TR-TCO-TCSS-TCP-TCI-TCS-TCU-TCLS. However, if we directly calculate equation with the results listed in Table 9, for example as the GA column, 184267.47-39-18-7856.75-246.46-3216.75-426.64-1625.01=170838.86 instead of the result 100538.86 listed in Table 9. In addition, as the authors have changed the equation (2), the result should be accordingly updated.

Reviewer #2: Most issues have been resolved in the revised manuscript. If you were not already part of the journal's reviewer pool, you have now been added to it. We look forward to your continued participation in our journal, and we hope you will consider us again for future submissions.

7. PLOS authors have the option to publish the peer review history of their article (what does this mean?). If published, this will include your full peer review and any attached files.

Reviewer #1: No

Reviewer #2: No

---

## [Author Response · Author response to Decision Letter 1]

18 Jan 2022

Response to Reviewers' Comments on [PONE-D-21-28756R1] - [EMID: 6471bfd9f0373dab]

An integrated optimization model and metaheuristics for assortment planning, shelf space allocation, and inventory management of perishable products: A real application

We appreciate that the reviewer’s comments. The followings are our point-by-point responses:

Reviewer #1: The authors have already addressed some of my concerns. However, for the question 4 about the inconsistence of the result in Tables 6, 7 and 9 still makes me confused. Taking Table 9 as an example, according to the equation (1), TP=TR-TCO-TCSS-TCP-TCI-TCS-TCU-TCLS. However, if we directly calculate equation with the results listed in Table 9, for example as the GA column, 184267.47-39-18-7856.75-246.46-3216.75-426.64-1625.01=170838.86 instead of the result 100538.86 listed in Table 9. In addition, as the authors have changed the equation (2), the result should be accordingly updated.

Thanks 

1) Many thanks for bringing this issue into our attention. Yes, that’s correct. We check all data and table numbers and should say with a great apology, we have Typo mistakes in table numbers, and some digit has missed in typing.

We checked entire numeric results in all tables and found some mistakes. We revised these mistakes as follows:

1274.3 instead of 127.3 in table 6 (digit 4 was missed), 

7769.028 instead of 769.028 in table 7(digit 7 was missed), 

78156.75 instead of 7856.75 in table 9 (digit 1 was missed)

2) Thank you very much for your thoughtful comment. After this comment, we have made an attempt to update results. To this end, we run the model and found that the results are same. After a detailed review, we found that two equations are working in the same manner.

In first version, the equation 2 wasTR=∑_i▒∑_k▒∑_g▒∑_t▒〖p_i.〖(m〗_ikt^g)〗, in which m_ikt^G Represents the amount of product i with lifetime g that is substituted by the product k at time t. If i=k, this case represents direct sale of product i for demand of product i.

In second version (Revision 1), to clear up the ambiguity that occurred to the referee we clarify this equation to 

TR=∑_i▒∑_(k≠i)▒∑_g▒∑_t▒〖p_i.〖(m〗_ikt^g+x_i^t)〗

While i≠k. Consequently, it should be noticed that the results of these equations are the same and in case i=k, 〖∑_k▒∑_g▒m_ikt^g =x〗_i^t.

A summary of comparison is given in the table below 

 1 2

Eq. TR=∑_i▒∑_k▒∑_g▒∑_t▒〖p_i.〖(m〗_ikt^g)〗 TR=∑_i▒∑_(k≠i)▒∑_g▒∑_t▒〖p_i.〖(m〗_ikt^g+x_i^t)〗

substitution matrix w_ik=■(1&w_12&w_13@w_21&1&w_23@w_31&w_32&1) w_ik=■(0&w_12&w_13@w_21&0&w_23@w_31&w_32&0)

In result, the first version is considered in this revised paper which are able to consider direct sale.

---

## [Decision Letter · Decision Letter 2]

7 Feb 2022

An integrated optimization model and metaheuristics for assortment planning, shelf space allocation, and inventory management of perishable products: A real application

PONE-D-21-28756R2

Dear Dr. Ahmadi,

We’re pleased to inform you that your manuscript has been judged scientifically suitable for publication and will be formally accepted for publication once it meets all outstanding technical requirements.

Kind regards,

Yong Wang

Academic Editor

PLOS ONE

Additional Editor Comments (optional):

Reviewers' comments:

Reviewer's Responses to Questions

**Comments to the Author**

1. If the authors have adequately addressed your comments raised in a previous round of review and you feel that this manuscript is now acceptable for publication, you may indicate that here to bypass the “Comments to the Author” section, enter your conflict of interest statement in the “Confidential to Editor” section, and submit your "Accept" recommendation.

Reviewer #1: All comments have been addressed

2. Is the manuscript technically sound, and do the data support the conclusions?

Reviewer #1: Yes

3. Has the statistical analysis been performed appropriately and rigorously? 

Reviewer #1: Yes

4. Have the authors made all data underlying the findings in their manuscript fully available?

Reviewer #1: Yes

5. Is the manuscript presented in an intelligible fashion and written in standard English?

Reviewer #1: Yes

6. Review Comments to the Author

Reviewer #1: Thank you for the authors' effort in revising the paper. The authors have addressed all of my concerns.

7. PLOS authors have the option to publish the peer review history of their article (what does this mean?). If published, this will include your full peer review and any attached files.

Reviewer #1: No

---

## [Editor Report · Acceptance letter]

10 Feb 2022

PONE-D-21-28756R2 

An integrated optimization model and metaheuristics for assortment planning, shelf space allocation, and inventory management of perishable products: A real application 

Dear Dr. Ahmadi:

I'm pleased to inform you that your manuscript has been deemed suitable for publication in PLOS ONE. Congratulations! Your manuscript is now with our production department. 

Kind regards, 

on behalf of

Dr. Yong Wang 

Academic Editor

PLOS ONE